# Multiplicative Logit Adjustment Approximates Neural-Collapse-Aware Decision Boundary Adjustment

**Naoya Hasegawa & Issei Sato**
The University of Tokyo
{hasegawa-naoya410, sato}@g.ecc.u-tokyo.ac.jp

## Abstract

Real-world data distributions are often highly skewed. This has spurred a growing body of research on long-tailed recognition, aimed at addressing the imbalance in training classification models. Among the methods studied, multiplicative logit adjustment (MLA) stands out as a simple and effective method. What theoretical foundation explains the effectiveness of this heuristic method? We provide a justification for the effectiveness of MLA with the following two-step process. First, we develop a theory that adjusts optimal decision boundaries by estimating feature spread on the basis of neural collapse. Second, we demonstrate that MLA approximates this optimal method. Additionally, through experiments on long-tailed datasets, we illustrate the practical usefulness of MLA under more realistic conditions. We also offer experimental insights to guide the tuning of MLA hyperparameters.

## 1 Introduction

Publicly available benchmark datasets commonly used to evaluate classification models, such as MNIST (Lecun et al., 1998) and CIFAR100 (Krizhevsky, 2009), usually have a balanced number of samples per class. In contrast to benchmark datasets, empirical evidence indicates that real-world data often follow an exponential distribution (Reed, 2001). This also holds for classification problems involving real-world data (Spain & Perona, 2007; Li et al., 2017; 2022a). Such distributions are commonly referred to as long-tailed data. In long-tailed data, a few classes (head classes) have a large number of samples, while most classes (tail classes) have only a limited number of samples. This imbalance poses a significant challenge in classification tasks, known as long-tailed recognition (LTR). LTR focuses on improving the accuracy of models trained on long-tailed data when evaluated on uniformly distributed data. In LTR, there are many classes, and model predictions are often biased toward head classes. Since tail classes make up the majority, this bias significantly reduces the overall average accuracy across all classes (Zhang et al., 2021; Yang et al., 2022a).

Various methods have been proposed for LTR. State-of-the-art methods (Ma et al., 2021; Long et al., 2022; Tian et al., 2022) are generally complex but often consist of combinations of simpler techniques, such as resampling (Drummond & Holte, 2003) and two-stage training (Kang et al., 2020). One straightforward method is logit adjustment (LA) (Menon et al., 2020; Kim & Kim, 2020). Among these techniques, post-hoc LA is a technique that adjusts the linear classifier on top of the feature map without additional training after the standard training process, offering both effectiveness and simplicity. A prominent example of post-hoc LA is additive LA (ALA) (Menon et al., 2020), which is based on Fisher-consistent loss. Another type of post-hoc LA is multiplicative LA (MLA) (Kim & Kim, 2020), which aims to adjust the decision boundaries on the basis of the differences in the feature spread across classes. Since MLA has also demonstrated significant empirical success (Hasegawa & Sato, 2023), it raises the following question.

> Is there a theoretical foundation behind the effectiveness of MLA?

**Contribution**   We provide a theoretical guarantee that MLA achieves near-optimal decision boundary adjustments. Our contributions are summarized as follows.

- We derive a theoretical framework for adjusting decision boundaries optimally by using feature spread estimates from neural collapse (NC) (Papyan et al., 2020) (Section 4.2).

- We demonstrate that MLA is effective for LTR by proving it has similar decision boundaries to the method based on the aforementioned theory (Section 4.3). This clarifies under what conditions this approximation holds and how adjustments should be made.

- We experimentally validate that the approximation of MLA holds under realistic, non-ideal conditions where NC is not fully realized (Section 5).

- We provide empirical guidelines for hyperparameter-tuning of MLA (Section 5.4).

## 2 RELATED WORK

**Long-Tailed Recognition** LTR methods fall into three types: "Information Augmentation", "Module Improvement", and "Class Re-balancing" (Zhang et al., 2021). "Information Augmentation" aims to mitigate accuracy degradation by supplementing the limited information available for tail classes (Liu et al., 2020; Chu et al., 2020; Wang et al., 2021; Li et al., 2022b; Wang et al., 2023a). "Module Improvement" focuses on enhancing individual components of the network to boost overall performance. For example, Kang et al. (2020) developed a two-stage training method that separates the training of feature maps and classifiers. Yang et al. (2022b) and Liu et al. (2023) proposed methods to promote NC in feature maps. Recent approaches also use contrastive learning (Hadsell et al., 2006) (Ma et al., 2021; Tian et al., 2022; Li et al., 2022c; Wang et al., 2022; Kang et al., 2023) and vision-language models such as CLIP (Radford et al., 2021) (Ma et al., 2021; Tian et al., 2022; Long et al., 2022). Despite these advancements, managing these methods can be challenging due to slow convergence (Liu et al., 2023) or complex models (Hasegawa & Sato, 2023). "Class Re-balancing" strategies adjust class imbalance to prevent accuracy degradation. Techniques in this category include resampling (Drummond & Holte, 2003; Li et al., 2022b) and loss reweighting (Cui et al., 2019; Ma et al., 2022). However, reweighting is known to be ineffective for training overparameterized models (Byrd & Lipton, 2019; Zhai et al., 2022). LA is a simple yet effective method that falls under this category.

**Logit Adjustment** LA modifies logit values by using the number of samples in each class, to bridge the distribution gap between the training and test data. LA methods fall into two categories: loss-function adjustments (Cao et al., 2019; Menon et al., 2020; Tang et al., 2020; Kini et al., 2021; Wang et al., 2023b) and post-hoc LA methods that do not require additional training (Menon et al., 2020; Kim & Kim, 2020). The latter is particularly advantageous as it eliminates the need for retraining during hyperparameter tuning, thereby reducing both effort and time. Consequently, this paper delves deeper into post-hoc LA. Among post-hoc methods, there are two types depending on how the logits are adjusted: ALA (Menon et al., 2020) and MLA (Kim & Kim, 2020). ALA is grounded in a Fisher-consistent loss (Menon et al., 2020), providing some theoretical justification for its method. However, ALA assumes an infinite number of training samples, raising questions about its effectiveness with finite sample sizes. In contrast, MLA intuitively adjusts decision boundaries by using the size of each class cluster (Kim & Kim, 2020). MLA has empirically outperformed ALA in certain cases (Hasegawa & Sato, 2023).

**Neural Collapse** NC is a term introduced by Papyan et al. (2020) to describe four phenomena observed in the terminal phase of training classification models. These phenomena are as follows.

NC1: Training feature vectors within each class converge to their respective class means.

NC2: The class means of training feature vectors converge to form a simplex Equiangular Tight Frame (ETF) (Strohmer & Heath, 2003).

NC3: The class means of the training feature vectors align with the corresponding weight vectors of the linear classifier.

NC4: In prediction, the model outputs the class whose mean training feature vector is closest to the input feature vector in Euclidean distance.

NC is known to improve generalization accuracy and robustness (Papyan et al., 2020). Additionally, Galanti et al. (2021) theoretically demonstrated that NC can occur even with test samples or

samples from unseen classes. Various studies have explored the conditions under which NC arises (Rangamani & Banburski-Fahey, 2022; Han et al., 2022), and it is known that models trained using cross-entropy loss can also exhibit NC (Lu & Steinerberger, 2021; Ji et al., 2022). However, it has been observed that training with imbalanced data may hinder NC (Fang et al., 2021; Thrampoulidis et al., 2022; Dang et al., 2024). To address this problem, some methods have been proposed to promote NC even with imbalanced data (Yang et al., 2022b; Liu et al., 2023). Yang et al. (2022b) introduced the ETF classifier, which fixes the linear classifier weights to form an ETF, thereby encouraging NC.

## 3 PRELIMINARIES

We outline the notations used throughout this paper. For a comprehensive list, refer to the table in Appendix A. For any integer $i$, $[i]$ represents the set $\{1, \ldots, i\}$. We use $\mathcal{X} \subset \mathbb{R}^p$ to denote the instance space and $\mathcal{Y} \equiv [K]$ to denote the label space. In a $K$-class classification problem, each sample $(\boldsymbol{x}, y) \in \mathcal{X} \times \mathcal{Y}$ is drawn from a distribution $P$. The training dataset, $\mathcal{S} \equiv \{(\boldsymbol{x}_i, y_i) \mid \boldsymbol{x}_i \in \mathcal{X}, y_i \in \mathcal{Y}\}_{i=1}^N$, consists of independent and identically distributed (i.i.d.) samples drawn from $P$. Partitioning by class, we define $\mathcal{S}_k \equiv \{\boldsymbol{x}_i \mid (\boldsymbol{x}_i, y_i) \in \mathcal{S}, y_i = k\}$ as the set of samples drawn from the class-conditional distribution $P_k$. Let $N \equiv |\mathcal{S}|$ be the total number of samples, and $n_k = |\mathcal{S}_k|$ be the number of samples in class $k$. Without loss of generality, we assume the classes are sorted in descending order of sample size, i.e., $n_k \geq n_{k+1}$ for all $k \in [K-1]$. The imbalance factor is defined by $\rho \equiv \frac{n_1}{n_K} = \frac{\max_k n_k}{\min_k n_k}$, which measures the degree of imbalance in the training dataset, where $\rho \gg 1$ holds in LTR scenarios. In contrast to the imbalanced training dataset, the test dataset used for evaluation has a uniform label distribution. Specifically, we denote $\tilde{\mathcal{S}}_k \sim P_k^{\tilde{n}_k}$ by the test dataset for class $k$, with $\tilde{n}_k$ samples. Then, it holds that $\tilde{n}_k = \tilde{n}_{k'}$ for all $k, k' \in \mathcal{Y}$.

Let $\boldsymbol{f} : \mathbb{R}^p \to \mathbb{R}^d$ be a feature map from the set $\mathcal{F} \subset \{\boldsymbol{f}' : \mathbb{R}^p \to \mathbb{R}^d\}$. We focus on the input to the final layer of this feature map. Suppose the features are given by $\boldsymbol{f}(\boldsymbol{x}) = \boldsymbol{W}^1 \boldsymbol{h}(\boldsymbol{x})$, where $\boldsymbol{W}^1 \in \mathbb{R}^{d \times d_1}$ is a linear layer, and $\boldsymbol{h} \in \mathcal{H} : \mathbb{R}^p \to \mathbb{R}^{d_1}$ represents the output of the second-to-last layer of the feature map. For simplicity, we assume that $\boldsymbol{h}$ is bounded, meaning there exists a constant $B \geq 1$ such that $\sup_{\boldsymbol{x} \in \mathcal{X}, \boldsymbol{h} \in \mathcal{H}} \|\boldsymbol{h}(\boldsymbol{x})\| \leq B$, where $\|\cdot\|$ denotes the Euclidean norm for vectors.

Let the expected value and mean of the features be denoted as $\boldsymbol{\mu_f}(P_k) \equiv \mathbb{E}_{\boldsymbol{x} \sim P_k}[\boldsymbol{f}(\boldsymbol{x})]$ and $\boldsymbol{\mu_f}(\mathcal{S}_k) \equiv \frac{1}{n_k} \sum_{\boldsymbol{x} \in \mathcal{S}_k} \boldsymbol{f}(\boldsymbol{x})$, respectively. When $\boldsymbol{f}$ is clear from the context, we abbreviate these to $\boldsymbol{\mu}(P_k)$ and $\boldsymbol{\mu}(\mathcal{S}_k)$.

These features are fed into a linear classifier $\boldsymbol{W} \in \mathbb{R}^{d \times K}$ and we obtain the logits $\boldsymbol{g}(\boldsymbol{x}) \equiv \boldsymbol{W}^\top \boldsymbol{f}(\boldsymbol{x})$, where $\cdot^\top$ denotes the transpose of a matrix. The linear classifier weights can be expressed using column vectors $\boldsymbol{w}_k \in \mathbb{R}^d$, where $\boldsymbol{W} = [\boldsymbol{w}_1, \ldots, \boldsymbol{w}_K]$. Thus, the logit for class $k$ can be written as $g_k(\boldsymbol{x}) \equiv \boldsymbol{w}_k^\top \boldsymbol{f}(\boldsymbol{x})$. We assume that the weights of the linear classifier form an ETF. This assumption holds under NC and when we use an ETF classifier (Yang et al., 2022b). This leads to the following equation:

$$\boldsymbol{w}_k^\top \boldsymbol{w}_{k'} = \begin{cases} 1 & \text{if } k = k', \\ -\frac{1}{K-1} & \text{otherwise.} \end{cases} \tag{1}$$

**Logit Adjustment** MLA is a method that adjusts logits by scaling the norm of the linear classifier weights for each class (Kim & Kim, 2020). This is equivalent[1] in classification outcome to adjusting the logit for class $k$, $g_k(\boldsymbol{x})$, by scaling it as $n_k^{-\gamma_\times} g_k(\boldsymbol{x})$. Here, $\gamma_\times > 0$ is a hyperparameter. Kim & Kim (2020) used projected gradient descent to normalize the norm of the linear layer, $\|\boldsymbol{w}_k\|$, to 1. This normalization step is unnecessary in our setting because we use an ETF classifier and the weights remain fixed. In contrast, ALA adjusts logits additively. This is equivalent[1] to modifying the logit for class $k$, $g_k(\boldsymbol{x})$, by subtracting $\gamma_+ \log n_k$, where $\gamma_+ > 0$ is a hyperparameter.

---

[1]Kim & Kim (2020) adjusted logits to $\left(\frac{n_1}{n_k}\right)^{\gamma_\times} g_k(\boldsymbol{x})$, while Menon et al. (2020) adjusted them to $g_k(\boldsymbol{x}) - \gamma_+ \log \frac{n_k}{\sum_{k'} n_{k'}}$. Note that both methods differ from the adjustment used in this paper only by a constant multiple or constant term. However, since this does not affect the ranking of logits across classes or the classification outcomes, we adopt the adjustments described in the main text of this paper.

## 4 MAIN ANALYSIS

MLA is a post-hoc adjustment method that increases the probability of classifying samples into tail classes by changing the angles of the decision boundaries (Kim & Kim, 2020). To derive the appropriate angles, it is necessary to estimate the sizes of the feature spread for each class. Galanti et al. (2021) demonstrated that NC also occurs in test samples and provided a quantitative measure of feature spread. Following their approach, we verify the effectiveness of MLA theoretically through the lens of NC. In the following sections, we employ the ETF classifier (Yang et al., 2022b) to encourage NC. This assumption can also be derived from the assumption that NC2 and NC3 hold. Using an ETF classifier also ensures that the weight vectors are normalized, as shown to be effective by Kim & Kim (2020).

First, in Section 4.1 we outline the assumptions and additional notations necessary to present our theory. Next, in Section 4.2, we demonstrate how to optimally adjust the decision boundaries from the perspective of NC. In Section 4.3, we indicate that MLA approximates this adjustment. Finally, Section 4.4 compares ALA and MLA in view of our theory.

### 4.1 PRELIMINARIES: SVD AND RADEMACHER COMPLEXITY

We consider the singular value decomposition of $\boldsymbol{W}^1$. When $\boldsymbol{W}^1$ has rank $r$, we can express it as $\boldsymbol{W}^1 = \sum_{l=1}^{r} s_l \boldsymbol{u}_l \boldsymbol{v}_l^\top$, where for each $l \in [r]$, $\boldsymbol{u}_l \in \mathbb{R}^d$, $\boldsymbol{v}_l \in \mathbb{R}^{d_1}$, and $s_l \in \mathbb{R}_{>0}$. Moreover, for all $l \in [r]$, we have $\|\boldsymbol{v}_l\| = \|\boldsymbol{u}_l\| = 1$, and for $l \neq l' \in [r]$, $\boldsymbol{v}_l^\top \boldsymbol{v}_{l'} = \boldsymbol{u}_l^\top \boldsymbol{u}_{l'} = 0$. Without loss of generality, we assume that $\forall l \in [r-1]$, $s_l \geq s_{l+1}$. In many deep neural networks, weight matrices can be replaced with low-rank representations while maintaining performance (Frankle & Carbin, 2018; Idelbayev & Carreira-Perpiñán, 2020; Kajitsuka & Sato, 2023). Thus, it is reasonable to consider $r \ll \min(d, d_1)$ in practical situations.

We define the set $\mathcal{H}_l$ as:

$$\mathcal{H}_l \equiv \{(\boldsymbol{v}_l, \boldsymbol{h}) \mid \boldsymbol{f} \in \mathcal{F}, \boldsymbol{f} = \boldsymbol{W}^1 \boldsymbol{h}, \; \boldsymbol{v}_l \text{ is the } l\text{-th right singular vector of } \boldsymbol{W}^1\}. \quad (2)$$

To evaluate the generalization performance of $\mathcal{H}_l$, we use the Rademacher complexity. The Rademacher complexity and empirical Rademacher complexity of $\mathcal{H}_l$ are defined by:

$$\mathfrak{R}_{n_k}(\mathcal{H}_l) \equiv \mathbb{E}_{\mathcal{S}_k' \sim P_k^{n_k}, \boldsymbol{\sigma}} \left[ \sup_{(\boldsymbol{v}_l, \boldsymbol{h}) \in \mathcal{H}_l} \frac{1}{n_k} \sum_{\boldsymbol{x}_j \in \mathcal{S}_k'} \sigma_j \boldsymbol{v}_l^\top \boldsymbol{h}(\boldsymbol{x}_j) \right], \quad (3)$$

$$\hat{\mathfrak{R}}_{\mathcal{S}_k}(\mathcal{H}_l) \equiv \mathbb{E}_{\boldsymbol{\sigma}} \left[ \sup_{(\boldsymbol{v}_l, \boldsymbol{h}) \in \mathcal{H}_l} \frac{1}{n_k} \sum_{\boldsymbol{x}_j \in \mathcal{S}_k} \sigma_j \boldsymbol{v}_l^\top \boldsymbol{h}(\boldsymbol{x}_j) \right], \quad (4)$$

where $\boldsymbol{\sigma} = [\sigma_1, \ldots, \sigma_{n_k}]^\top$ represents the i.i.d. Rademacher random variables. In many cases, the Rademacher complexity scales as $\mathfrak{R}_{n_k}(\mathcal{H}_l) \leq \frac{\mathcal{C}(\mathcal{H}_l, \mathcal{X})}{\sqrt{n_k}}$, where $\mathcal{C}(\mathcal{H}_l, \mathcal{X})$ is some complexity measure independent of $n_k$ (Cao et al., 2019; Galanti et al., 2021). For instance, this holds for rectified linear unit (ReLU) neural networks; see Appendix C.2 for further details. We make this assumption in our settings. We define the mean of the complexity $\bar{\mathcal{C}}(\mathcal{F}, \mathcal{X}) \equiv \frac{1}{r} \sum_{l=1}^{r} \mathcal{C}(\mathcal{H}_l, \mathcal{X})$

### 4.2 OPTIMAL DECISION BOUNDARY ADJUSTMENT FROM THE PERSPECTIVE OF NC

We indicate that the following result holds for general neural networks. For the proofs of the propositions presented in this section and for the case of ReLU neural networks, see Appendix C.

In NC, the training features converge to their respective class means (Papyan et al., 2020). It has also been demonstrated that test features are within a certain range from the class mean of training features, with this range depending on the sample size (Galanti et al., 2021). Following a similar approach, we can quantitatively measure the angular deviation between test features and the corresponding training class mean, as a function of the sample size. In LTR, sample sizes vary significantly across classes, resulting in a considerable imbalance in the angular deviation. Thus, we aim to adjust the decision boundary accordingly.

To achieve this, we first define the angular bound probability $\Pi(\theta; k)$ as follows.

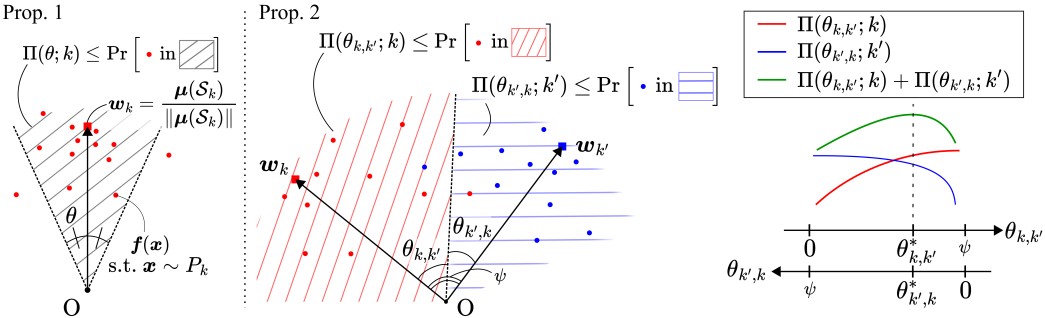

Figure 1: Overview of Propositions 1 and 2. The angular bound probability $\Pi(\theta; k)$ represents the lower bound of the probability that the feature vector of $\boldsymbol{x}$ sampled from $P_k$ lies within the shaded region. Proposition 1 indicates that $\Pi(\theta; k) = 1 - \tilde{\mathcal{O}}\left(1/\sqrt{n_k}\right)$. Proposition 2 offers the optimal decision boundary by maximizing $\Pi(\theta_{k,k'}; k) + \Pi(\theta_{k',k}; k')$ with respect to $\theta_{k,k'}$ and $\theta_{k',k}$.

**Definition 1** (Angular bound probability). *The angular bound probability $\Pi(\theta; k)$ for class $k$ is the lower bound of the probability that the angle between $\boldsymbol{f}(\boldsymbol{x})$ and $\boldsymbol{\mu}(\mathcal{S}_k)$, for $\boldsymbol{x} \sim P_k$, is less than $\theta$.*

That is, we have that $\Pi(\theta; k) \leq \Pr_{\boldsymbol{x} \sim P_k}[\angle(\boldsymbol{f}(\boldsymbol{x}), \boldsymbol{\mu}(\mathcal{S}_k)) < \theta]$, where $\angle(\cdot, \cdot)$ is the angle between the two vectors. The angular bound probability can also be seen as concentration of the class $k$ features. The following proposition provides a quantitative measure of $\Pi(\theta; k)$:

**Proposition 1.** *Suppose $n_k > 2$. For $\boldsymbol{f} \in \mathcal{F}$, assume that for all $\boldsymbol{x} \in \mathcal{S}_k$, $\boldsymbol{f}(\boldsymbol{x}) = \boldsymbol{\mu}(\mathcal{S}_k)$ holds. Consider any $\theta$ that satisfies the following condition:*

$$\frac{\pi}{2} \frac{r\|\boldsymbol{W}^1\|_2}{\sqrt{n_k}\|\boldsymbol{\mu}(\mathcal{S}_k)\|} \left(4\bar{\mathcal{C}}(\mathcal{F}, \mathcal{X}) + 4B + B\sqrt{2\log \frac{\sqrt{n_k}\|\boldsymbol{\mu}(\mathcal{S}_k)\|}{\|\boldsymbol{W}^1\|_2}}\right) < \theta < \frac{\pi}{2}, \tag{5}$$

*where $\|\cdot\|_2$ denotes the spectral norm. For such $\theta$, the following holds:*

$$\Pi(\theta; k) = 1 - \frac{\pi}{2\theta} \frac{r\|\boldsymbol{W}^1\|_2}{\sqrt{n_k}\|\boldsymbol{\mu}(\mathcal{S}_k)\|} \left(4\bar{\mathcal{C}}(\mathcal{F}, \mathcal{X}) + 4B + 1 + B\sqrt{2\log \frac{2\theta\sqrt{n_k}\|\boldsymbol{\mu}(\mathcal{S}_k)\|}{\pi\|\boldsymbol{W}^1\|_2}}\right). \tag{6}$$

The overview of this proposition is illustrated in the left side of Figure 1. When NC occurs, note that $\boldsymbol{f}(\boldsymbol{x}) = \boldsymbol{\mu}(\mathcal{S}_k)$ holds for all $\boldsymbol{x} \in \mathcal{S}_k$ by NC1 and $\boldsymbol{w}_k = \frac{\boldsymbol{\mu}(\mathcal{S}_k)}{\|\boldsymbol{\mu}(\mathcal{S}_k)\|}$ holds by NC3. Additionally, observe that the left-hand side of Eq. (5) and $1 - \Pi(\theta; k)$ are $\tilde{\mathcal{O}}\left(1/\sqrt{n_k}\right)$, where $\tilde{\mathcal{O}}$ denotes asymptotic notation ignoring logarithmic terms. From this proposition, we can quantitatively infer that as the sample size of a class increases, the features become more concentrated within a narrower region, with higher probability. This allows us to consider the optimal angle for the decision boundary.

Consider the optimal angle for the decision boundary between two classes, $k$ and $k'$. Since we use an ETF classifier, the linear classifier weights forms an ETF structure, meaning the angle between $\boldsymbol{w}_k$ and $\boldsymbol{w}_{k'}$ is $\psi \equiv \arccos\left(-\frac{1}{K-1}\right)$. Now, consider a decision boundary that forms an angle $\theta_{k,k'}$ with $\boldsymbol{w}_k$. The angle between this boundary and $\boldsymbol{w}_{k'}$ is $\theta_{k',k} \equiv \psi - \theta_{k,k'}$. Here, $0 < \theta_{k,k'}, \theta_{k',k} < \psi$.

In this setup, the probability that a sample $\boldsymbol{x} \sim P_k$ lies on the $\boldsymbol{w}_k$ side of the boundary is at least $\Pi(\theta_{k,k'}; k)$. Similarly, the probability that a sample $\boldsymbol{x} \sim P_{k'}$ lies on the $\boldsymbol{w}_{k'}$ side is at least $\Pi(\theta_{k',k}; k')$. Therefore, the accuracy of classification between the two classes is at least $\frac{1}{2}\left(\Pi(\theta_{k,k'}; k) + \Pi(\theta_{k',k}; k')\right)$, which depends on the angle $\theta_{k,k'}$ of the decision boundary. Thus, we seek the value of $\theta_{k,k'}$ that maximizes this lower bound. The following proposition provides the solution to this problem.

**Proposition 2.** *Suppose $n_k, n_{k'} > 2$. For $\boldsymbol{f} \in \mathcal{F}$, assume that for both $\hat{k} \in \{k, k'\}$, $\boldsymbol{f}(\boldsymbol{x}) = \boldsymbol{\mu}(\mathcal{S}_{\hat{k}})$ holds for all $\boldsymbol{x} \in \mathcal{S}_{\hat{k}}$. Consider the following maximization problem:*

$$\max_{\theta_{k,k'}, \theta_{k',k}} \Pi(\theta_{k,k'}; k) + \Pi(\theta_{k',k}; k') \quad \text{s.t.} \quad \theta_{k,k'}, \theta_{k',k} > 0, \quad \theta_{k,k'} + \theta_{k',k} = \psi. \tag{7}$$

*The unique solution $\theta_{k,k'}^*, \theta_{k',k}^*$ within the range where $\theta_{k,k'}$ and $\theta_{k',k}$ satisfy Eq. (5) is given by:*

$$\theta_{k,k'}^* = \psi \frac{\|\boldsymbol{\mu}(\mathcal{S}_{k'})\|\sqrt{n_{k'}}}{\|\boldsymbol{\mu}(\mathcal{S}_k)\|\sqrt{n_k} + \|\boldsymbol{\mu}(\mathcal{S}_{k'})\|\sqrt{n_{k'}}}, \quad \theta_{k',k}^* = \psi \frac{\|\boldsymbol{\mu}(\mathcal{S}_k)\|\sqrt{n_k}}{\|\boldsymbol{\mu}(\mathcal{S}_k)\|\sqrt{n_k} + \|\boldsymbol{\mu}(\mathcal{S}_{k'})\|\sqrt{n_{k'}}}. \tag{8}$$

The overview of this proposition is illustrated in the right side of Figure 1. When $n_k$ and $n_{k'}$ are sufficiently large, Eq. (5) holds over a wide range of $0 < \theta < \frac{\pi}{2}$. Moreover, as $K \to \infty$, $\psi = \arccos\left(-\frac{1}{K-1}\right) \to \frac{\pi}{2}$ and $\theta_{k,k'}^*, \theta_{k',k}^* < \frac{\pi}{2}$ holds, making Eq. (8) meet the condition of Eq. (5) when $K$ is sufficiently large. This is a reasonable assumption in LTR (Yang et al., 2022a). In practice, for CIFAR100-LT, $\psi - \frac{\pi}{2} \sim 0.01$, meaning that $\theta_{k,k'}^*, \theta_{k',k}^* < \frac{\pi}{2}$ is sufficiently valid.

This result can easily be extended to multi-class cases (refer to Proposition 3). In multi-class cases, it becomes an optimization problem of the angles $\theta_{k,k'}$ for every pair of classes $k, k' \neq k$ in $\mathcal{Y}$, defined as $\Theta \equiv \{\theta_{k,k'} \mid k \neq k' \in \mathcal{Y}\}$.

**Proposition 3.** *Suppose $n_k > 2$ for all $k \in \mathcal{Y}$. For $\boldsymbol{f} \in \mathcal{F}$, assume that for all $k \in \mathcal{Y}$, $\boldsymbol{f}(\boldsymbol{x}) = \boldsymbol{\mu}(\mathcal{S}_k)$ holds for all $\boldsymbol{x} \in \mathcal{S}_k$. Consider the following maximization problem:*

$$\max_{\Theta} \sum_{k \in \mathcal{Y}} \sum_{k' \neq k} \Pi(\theta_{k,k'}; k) + \Pi(\theta_{k',k}; k') \quad s.t. \ \forall k, k' \in \mathcal{Y}, \theta_{k,k'}, \theta_{k',k} > 0 \ and \ \theta_{k,k'} + \theta_{k',k} = \psi. \tag{9}$$

*The unique solution $\theta_{k,k'}^* \in \Theta^*$ within the range where all $\theta_{k,k'}^* \in \Theta^*$ satisfies Eq. (5) is given by:*

$$\theta_{k,k'}^* = \psi \frac{\|\boldsymbol{\mu}(\mathcal{S}_{k'})\|\sqrt{n_{k'}}}{\|\boldsymbol{\mu}(\mathcal{S}_k)\|\sqrt{n_k} + \|\boldsymbol{\mu}(\mathcal{S}_{k'})\|\sqrt{n_{k'}}}. \tag{10}$$

These theories suggest replacing the linear layer with a 1-vs-1 multi-class classifier with decision boundaries of $\Theta^*$. To generalize, we introduce a hyperparameter $\gamma_{1v1}$, and define $\theta_{k,k'}^*(\gamma_{1v1}) = \psi \frac{\|\boldsymbol{\mu}(\mathcal{S}_{k'})\|n_{k'}^{\gamma_{1v1}}}{\|\boldsymbol{\mu}(\mathcal{S}_k)\|n_k^{\gamma_{1v1}} + \|\boldsymbol{\mu}(\mathcal{S}_{k'})\|n_{k'}^{\gamma_{1v1}}}$, with $\Theta_{\gamma_{1v1}}^* \equiv \{\theta_{k,k'}^*(\gamma_{1v1}) \mid k \neq k' \in \mathcal{Y}\}$. Note that $\Theta_{\gamma_{1v1}}^* = \Theta^*$ holds when $\gamma_{1v1} = \frac{1}{2}$. We refer to the 1-vs-1 multi-class classifier that classifies based on this $\Theta_{\gamma_{1v1}}^*$ as 1vs1adjuster. See Appendix D for the specific algorithm of 1vs1adjuster.

## 4.3 MLA SOLVES LTR PROBLEMS WITH DECISION BOUNDARIES AKIN TO 1VS1ADJUSTER

We demonstrate that MLA addresses the LTR problem. Since Section 4.2 proves 1vs1adjuster has the optimal decision boundaries for LTR, we verify MLA operates similarly to 1vs1adjuster. MLA can be viewed as a technique that multiplicatively adjusts the cosine similarity between features and the linear layer weight vectors. This increases the probability of classifying samples into tail classes. Although our theory in Section 4.2 aims to find the optimal decision boundaries between classes, it can be connected to MLA by considering the magnitude of logits under simple assumptions.

For the following discussion, we assume that $K$ is sufficiently large. Under this condition, it holds that $\psi = \arccos\left(-\frac{1}{K-1}\right) \to \frac{\pi}{2}$, allowing us to assume that $\cos(\theta_{k,k'}^*(\gamma_{1v1})), \cos(\theta_{k',k}^*(\gamma_{1v1})) > 0$.

Let the MLA factor, multiplicatively applied to the logits of class $k$, be denoted by $\kappa_k$. In other words, we consider the case in which the logit $g_k(\boldsymbol{x})$ for class $k$ is adjusted to $\kappa_k g_k(\boldsymbol{x})$. We present that when adjusting the decision boundaries on the basis of 1vs1adjuster, $\kappa_k \propto n_k^{-\gamma_{1v1}}$ holds. Note that the ETF classifier ensures equal norms of the linear layer weight vectors across all classes. To adjust the decision boundaries to be equivalent to 1vs1adjuster by using MLA, we should adjust $\kappa_k$ such that the following holds for all $k \neq k' \in \mathcal{Y}$:

$$\kappa_k \cos(\theta_{k,k'}^*(\gamma_{1v1})) = \kappa_{k'} \cos(\theta_{k',k}^*(\gamma_{1v1})). \tag{11}$$

A problem arises here because there are only $K$ MLA factors, while they must satisfy $K(K-1)/2$ equations. Consequently, when $K > 2$, there is typically no solution. However, by considering the following approximation, it is possible to obtain an approximate solution. First, note that as $K \to \infty$, it holds that $\psi \to \frac{\pi}{2}$. Letting $\epsilon = \frac{\pi}{2} - \psi$, and considering the continuity of cos, we can approximate and solve Eq. (11) as follows:

$$\frac{\kappa_k}{\kappa_{k'}} = \tan(\theta_{k,k'}^*(\gamma_{1v1}))\cos(\epsilon) - \sin(\epsilon) \tag{12}$$

$$\sim \tan(\theta_{k,k'}^*(\gamma_{1v1})). \tag{13}$$

Now, let $\tau_{k,k'}(\gamma_{1\mathrm{v}1}) \equiv \frac{\|\boldsymbol{\mu}(\mathcal{S}_k)\|n_k^{\gamma_{1\mathrm{v}1}}}{\|\boldsymbol{\mu}(\mathcal{S}_{k'})\|n_{k'}^{\gamma_{1\mathrm{v}1}}}$, then we have:

$$\theta_{k,k'}^*(\gamma_{1\mathrm{v}1}) = \psi \frac{\|\boldsymbol{\mu}(\mathcal{S}_{k'})\|n_{k'}^{\gamma_{1\mathrm{v}1}}}{\|\boldsymbol{\mu}(\mathcal{S}_k)\|n_k^{\gamma_{1\mathrm{v}1}} + \|\boldsymbol{\mu}(\mathcal{S}_{k'})\|n_{k'}^{\gamma_{1\mathrm{v}1}}} = \psi \frac{1}{\tau_{k,k'}(\gamma_{1\mathrm{v}1}) + 1} \sim \frac{\pi}{2} \frac{1}{\tau_{k,k'}(\gamma_{1\mathrm{v}1}) + 1}. \tag{14}$$

Next, for $0 \le \theta < 1$, we approximate $\tan\left(\frac{\pi}{2}\theta\right)$ using the rational function $\phi(\theta) = \frac{\theta}{1-\theta}$. This approximation satisfies the following properties:

**Lemma 1.**

$$\phi(0) = \tan(0) = 0, \tag{15}$$

$$\phi\left(\frac{1}{2}\right) = \tan\left(\frac{\pi}{4}\right) = 1, \tag{16}$$

$$\lim_{\theta \to 1-0} \phi(\theta) = \lim_{\theta \to 1-0} \tan\left(\frac{\pi}{2}\theta\right) = +\infty. \tag{17}$$

**Lemma 2.** *Let $\theta \in [0, 1)$. Then, the following holds:*

$$\phi(\theta) \le \tan\left(\frac{\pi}{2}\theta\right) < \frac{\pi}{2}\phi(\theta) \qquad \left(0 \le \theta \le \frac{1}{2}\right), \tag{18}$$

$$\frac{2}{\pi}\phi(\theta) < \tan\left(\frac{\pi}{2}\theta\right) < \phi(\theta) \qquad \left(\frac{1}{2} < \theta < 1\right). \tag{19}$$

The proof of Lemma 1 is trivial and thus omitted. The proof of Lemma 2 can be found in Appendix C.3. These lemmas suggest that the approximation $\phi(\theta) = \frac{\theta}{1-\theta}$ for $\tan\left(\frac{\pi}{2}\theta\right)$ is particularly close when $\theta = \frac{1}{2}$ or $\tau_{k,k'}(\gamma_{1\mathrm{v}1}) = 1$, but it provides a global approximation. This is especially suitable for LTR settings, where $\tau_{k,k'}(\gamma_{1\mathrm{v}1})$ tends to deviate significantly from 1.

Using these results, we further approximate Eq. (13) as:

$$\frac{\kappa_k}{\kappa_{k'}} \sim \tan(\theta_{k,k'}^*(\gamma_{1\mathrm{v}1})) \sim \frac{\frac{2}{\pi}\theta_{k,k'}^*(\gamma_{1\mathrm{v}1})}{1 - \frac{2}{\pi}\theta_{k,k'}^*(\gamma_{1\mathrm{v}1})} \sim \frac{1}{\tau_{k,k'}(\gamma_{1\mathrm{v}1})}. \tag{20}$$

Therefore, if we set $\kappa_k = \frac{\alpha}{\|\boldsymbol{\mu}(\mathcal{S}_k)\|n_k^{\gamma_{1\mathrm{v}1}}}$ for any $\alpha \in \mathbb{R}_{>0}$, we can approximately satisfy Proposition 3. This corresponds to performing MLA with $\gamma_\times = \gamma_{1\mathrm{v}1}$, under the assumption of feature normalization.

## 4.4 DISCUSSION: COMPARISON OF MLA WITH ALA

In Section 4.3, we demonstrate that MLA is grounded in the theoretical framework of decision boundary adjustment developed in Section 4.2. We now turn our attention to ALA, presenting the issues that arise when applying the theory from Section 4.2 to ALA. This highlights the differences between MLA and ALA, and indicates the relative advantages of MLA.

ALA is inconsistent with the concept of optimal decision boundary adjustment as described in Proposition 3. This is because the decision boundaries adjusted by ALA changes depending on the norm of each feature, $\|\boldsymbol{f}(\boldsymbol{x})\|$. For instance, let's consider an instance $\boldsymbol{x}$ whose logit of class $k$ is represented as $g_k(\boldsymbol{x}) = \|\boldsymbol{f}(\boldsymbol{x})\| \cos(\angle(\boldsymbol{f}(\boldsymbol{x}), \boldsymbol{w}_k))$. The output of the classification label for $\boldsymbol{x}$ depends on the magnitude of $\|\boldsymbol{f}(\boldsymbol{x})\| \cos(\angle(\boldsymbol{f}(\boldsymbol{x}), \boldsymbol{w}_k)) - \gamma_+ \log n_k$ for each $k$. In other words, the classification result depends on both the angle $\angle(\boldsymbol{f}(\boldsymbol{x}), \boldsymbol{w}_k)$ and the norm $\|\boldsymbol{f}(\boldsymbol{x})\|$. This implies that the optimal decision boundaries during inference are affected by the norm of individual features. This dependency conflicts with Proposition 3, in which the optimal decision boundaries are independent of the norm of each feature.

On the other hand, suppose we assume that $\|\boldsymbol{f}(\boldsymbol{x})\|$ is equal for all $\boldsymbol{x} \in \mathcal{X}$, ensuring that the angles of the decision boundaries remain constant for each sample. Following the same reasoning as in Section 4.3, we can also approximate 1vs1adjuster by ALA. However, compared to MLA,

this approximation cannot hold globally when $\tau_{k,k'}(\gamma_{1v1})$ varies significantly. This inconsistency with LTR settings, in which $\tau_{k,k'}(\gamma_{1v1})$ can deviate considerably, makes ALA incompatible. See Appendix E for details.

These points are corroborated by the subsequent experimental results, which demonstrate the substantial differences between the properties of MLA and ALA.

## 5 EXPERIMENTS

We experimentally demonstrate that MLA is an appropriate approximation of 1vs1adjuster from two perspectives. First, we outline the experimental setup in Section 5.1. Then, we demonstrate that MLA and 1vs1adjuster share similar decision boundaries in Section 5.2 and yield equivalent classification accuracy in Section 5.3. Finally, we present insights into MLA hyperparameters in Section 5.4. Code is available at `https://github.com/HN410/MLA-Approximates-NCDBA`.

### 5.1 SETTINGS

Our experimental setup primarily follows Hasegawa & Sato (2023). We used CIFAR10, CIFAR100 (Krizhevsky, 2009), and ImageNet (Deng et al., 2009) as datasets. Following Cui et al. (2019) and Liu et al. (2019), we created long-tailed versions of these datasets, namely CIFAR10-LT, CIFAR100-LT, and ImageNet-LT, with an imbalance ratio of $\rho = 100$. To demonstrate that this method applies to general modal data, not just images, we also used the tabular dataset Helena (Guyon et al., 2019). We can also show the effectiveness of the method on real data using Helena because this is an inherently imbalanced dataset with $\rho \simeq 40$.

For the network architecture, we used ResNeXt50 (Xie et al., 2017) for ImageNet-LT and ResNet34 (He et al., 2016) for the other image datasets unless otherwise specified. For Helena, we used a multi-layer perceptron (MLP), following Kadra et al. (2021). We adopted the ETF classifier as the linear classifier to promote NC and fix the linear classifier weights. Thus, only the parameters of the feature map were trained. We used cross-entropy loss as the loss function and applied weight decay (Hanson & Pratt, 1989) and feature regularization as regularization techniques. Feature regularization was employed to promote NC (Hasegawa & Sato, 2023) and to prevent significant differences in the norms of the training features across classes, fulfilling the condition for MLA to approximate 1vs1adjuster. We tuned the hyperparameters $\gamma_{1v1}$, $\gamma_+$, and $\gamma_\times$ using validation datasets. For further detailed experimental settings, refer to Appendix F.1.

We examined the behavior of MLA, ALA, and 1vs1adjuster when applied post-hoc to models trained on the training datasets. Specifically, we aimed to assess whether MLA provides an effective approximation of 1vs1adjuster in terms of decision boundary adjustment and classification accuracy. ALA, another post-hoc LA method, was included as a benchmark for comparison.

### 5.2 DECISION BOUNDARY ANGLES

We investigate how the decision boundary angles between classes are adjusted. Similar to the analysis in Sections 4.3 and 4.4, MLA and ALA can be interpreted as methods for adjusting the angles of the decision boundaries between classes. Let $\theta_{k,k'}^\times$ and $\theta_{k,k'}^+$ represent the angles of the decision boundaries between classes $k$ and $k'$ derived from MLA and ALA, respectively.

For MLA, the following holds from Eq. (12):

$$\theta_{k,k'}^\times = \arctan\left(\frac{\left(\frac{n_{k'}^{\gamma_\times}}{n_k^{\gamma_\times}} + \sin\left(\frac{\pi}{2} - \psi\right)\right)}{\cos\left(\frac{\pi}{2} - \psi\right)}\right). \tag{21}$$

For ALA, we assume that $\|\boldsymbol{f}(\boldsymbol{x})\|$ is equal across all instances $\boldsymbol{x}$ and sufficiently large, denoted as $\|\boldsymbol{f}\|$. Then, from Eqs. (65) and (66), the following holds:

$$\theta_{k,k'}^+ = \frac{\psi}{2} - \arcsin\left(\frac{\gamma_+ \log\frac{n_k}{n_{k'}}}{2\|\boldsymbol{f}\|\sin\left(\frac{\psi}{2}\right)}\right). \tag{22}$$

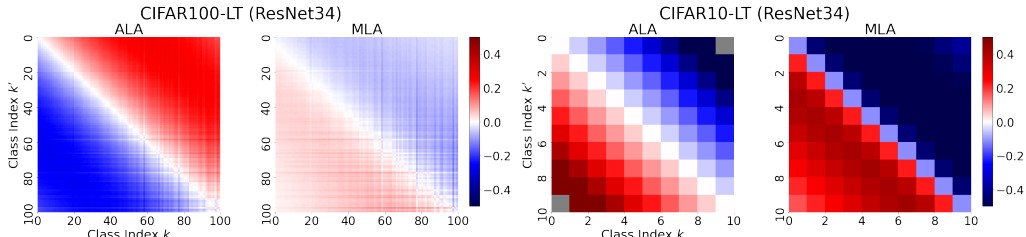

Figure 2: Heatmaps showing difference in angles of decision boundaries between each method and 1vs1adjuster. On the left is the result for CIFAR100-LT, and on the right is the result for CIFAR10-LT. The left side of each figure displays $\theta_{k,k'}^{+} - \theta_{k,k'}^{*}$, while the right side displays $\theta_{k,k'}^{\times} - \theta_{k,k'}^{*}$. Values that became NaN are shown in gray. In the case of CIFAR100-LT, the angle differences between MLA and 1vs1adjuster are generally small.

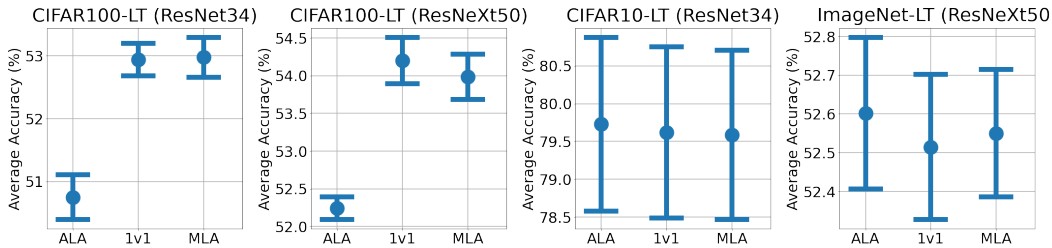

Figure 3: Average accuracy of each model trained on each dataset and adjusted by different methods. The error bars represent the mean and standard deviation across five trials with different seed values. 1v1 is short for 1vs1adjuster. MLA and 1vs1adjuster consistently achieve comparable accuracy.

We investigate the differences between these angles and $\theta_{k,k'}^{*}$ used in 1vs1adjuster and demonstrate that MLA serves as a sufficient approximation of 1vs1adjuster. Figure 2 displays heatmaps comparing $\theta_{k,k'}^{+} - \theta_{k,k'}^{*}$ and $\theta_{k,k'}^{\times} - \theta_{k,k'}^{*}$ for CIFAR100-LT and CIFAR10-LT. Detailed experimental settings and results for other datasets are provided in Appendices F.1 and F.2, respectively. The heatmaps representing the difference between MLA and 1vs1adjuster are generally faint on datasets with a sufficient number of classes, such as CIFAR100-LT. This means that $\theta_{k,k'}^{\times} - \theta_{k,k'}^{*}$ is globally small across all $k, k' \in \mathcal{Y}$. This is consistent with our theoretical predictions from Section 4.3, indicating that MLA is a sufficient approximation of 1vs1adjuster in large-class settings. The same phenomenon is observed for ImageNet-LT and Helena. Note that in these latter two datasets, this phenomenon occurs even though training accuracy is not sufficiently high and NC is not fully realized (see Table 1). This suggests that our theory holds under more relaxed conditions.

On CIFAR10-LT, which has fewer classes, the decision boundaries of both MLA and ALA differ significantly from those of 1vs1adjuster. This may be because the number of classes is so small that approximations such as $\psi \sim \frac{\pi}{2}$ are not valid. However, even in this case, MLA can still be regarded as an approximate method in that it achieves comparable test accuracy to 1vs1adjuster. See Section 5.3.

## 5.3 TEST ACCURACY

We compare the test accuracy when each method is applied post-hoc to trained models. Figure 3 displays the average test accuracy with error bars for each model trained on each dataset and adjusted by different methods. The results on Helena and the detailed accuracy table can be found in Appendix F.3. In all experiments, MLA can achieve high average accuracy comparable to 1vs1adjuster. Remarkably, this approximation holds even in cases where the assumptions mentioned in Section 4 do not hold, such as when the number of classes is small (CIFAR10-LT) or when training accuracy is low and NC has not sufficiently occurred (ImageNet-LT and Helena). In contrast, ALA achieves

Table 1: Optimal hyperparameters and training accuracy for each model and dataset. For models that achieve 100% training accuracy, $\gamma_\times^*$ and $\gamma_{1v1}^*$ are slightly higher than 0.5.

| Dataset | Model | $\gamma_\times^*$ | $\gamma_{1v1}^*$ | Training Accuracy (%) |
|---|---|---|---|---|
| CIFAR100-LT | ResNet34 | $0.710 \pm 0.037$ | $0.770 \pm 0.060$ | 100.0 |
| CIFAR100-LT | ResNeXt50 | $0.770 \pm 0.087$ | $1.010 \pm 0.058$ | 100.0 |
| CIFAR10-LT | ResNet34 | $0.780 \pm 0.518$ | $0.410 \pm 0.120$ | 100.0 |
| ImageNet-LT | ResNeXt50 | $0.200 \pm 0.000$ | $0.250 \pm 0.000$ | 86.6 |
| Helena | MLP | $0.480 \pm 0.093$ | $0.580 \pm 0.060$ | 42.1 |

lower accuracy than 1vs1adjuster in some cases. ALA is based on Fisher consistency (Menon et al., 2020), assuming that the number of training samples is sufficiently large. While our theory also assumes a certain level of training sample size, it has been shown to work well even with more practical sample sizes. In addition, MLA and 1vs1adjuster achieve comparable accuracy to ALA even under unfavorable conditions, such as experiments on CIFAR10-LT and ImageNet-LT, where the assumptions of our theory do not strictly hold. These results demonstrate that MLA is a robust and practical method for improving test accuracy, even in real-world scenarios where ideal theoretical conditions are not met.

### 5.4 Hyperparameters

Table 1 summarizes the optimal values of MLA and 1vs1adjuster hyperparameters, $\gamma_\times^*$ and $\gamma_{1v1}^*$, and the training accuracy for models trained on each training dataset.

On CIFAR10-LT, which has a small number of classes, the values of $\gamma_\times^*$ and $\gamma_{1v1}^*$ are significantly different. This is probably because the approximation of 1vs1adjuster to MLA does not hold well when the number of classes is small.

From here on, we focus on cases in which the number of classes is sufficiently large. On CIFAR100-LT, where training accuracy has reached $100\%$, the optimal value is slightly higher than the theoretically derived value of 0.5. This suggests that when training reaches terminal phase and NC occurs, the feature spread can be bounded to a smaller order than $\tilde{\mathcal{O}}\big(1/\sqrt{n_k}\big)$. For instance, recent research has aimed to bound generalization error to a smaller order than $\tilde{\mathcal{O}}\big(1/\sqrt{n_k}\big)$ (Wei & Ma, 2019), and this may be related.

On Helena and ImageNet-LT, where training accuracy has not reached $100\%$, the optimal values can be smaller than 0.5. This is likely because the feature spread between classes does not differ as much compared to when NC sufficiently occurs. Even in such cases, MLA and 1vs1adjuster can achieve high test accuracy by adjusting hyperparameters (see Section 5.3 and Appendix F.3). These results are expected to be key insights when searching for the optimal hyperparameters for MLA.

## 6 Conclusion

We provide a solid theoretical foundation for MLA, a simple heuristic method in LTR, and demonstrate its utility. First, on the basis of NC, we estimate the feature spread for each class and present a theory that derives the optimal method for adjusting the decision boundaries. Furthermore, we indicate that MLA is an approximation of this method and theoretically guarantees improvements in test accuracy. This provides clear insights into the conditions under which MLA is effective and how logits should be adjusted quantitatively. Additionally, through experiments, we demonstrate that the approximation holds under more relaxed conditions and that MLA exhibits practical usefulness by achieving accuracy comparable to or better than ALA. This research forms a crucial foundation for future advancements in LTR and imbalanced learning methods.

One limitation of this study is the lack of a more in-depth theoretical analysis of ALA. While the current theory does not align with ALA, there may be approaches or assumptions that could guarantee its performance with a practical number of samples. Future research could explore this direction. Another potential avenue is the combination of MLA with other methods to develop more sophisticated methods and achieve state-of-the-art improvements in accuracy.

ACKNOWLEDGMENTS

We extend our heartfelt gratitude to everyone who supported us in conducting this research.

We would like to thank the students of our lab and our fellow graduate friends for their insightful discussions, which greatly contributed to shaping the direction and resolving the challenges of this research. Their enthusiastic debates were crucial to the progress of this work.

We also express our deep gratitude to the reviewers of this paper. Their detailed and constructive comments significantly improved the structure and logical development of the paper, greatly enhancing the quality of this research.

This work was supported by JSPS KAKENHI Grant Number JP24H00709.

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

# A  ACRONYM AND NOTATION TABLE

Table 2 lists the acronyms referenced throughout this paper, and Tables 3, 4, and 5 provide a summary of the notations used in this paper.

Table 2: Acronym table

| Abbreviation | Definition |
|---|---|
| 1v1 | 1vs1adjuster |
| ALA | Additive logit adjustment (Menon et al., 2020) |
| ETF | Equiangular tight frame |
| i.i.d | Independent and identically distributed |
| LA | Logit adjustment |
| LTR | Long-tailed recognition |
| MLA | Multiplicative logit adjustment (Kim & Kim, 2020) |
| MLP | Multi-layer perceptron |
| NC | Neural collapse (Papyan et al., 2020) |
| ReLU | Rectified linear unit |

Table 3: Notation table for operators and set

| | |
|---|---|
| $\angle$ | The angle between the two vectors |
| $\|\cdot\|$ | The Euclidean norm for vectors |
| $\|\cdot\|_2, \|\cdot\|_F$ | The spectral norm and Frobenius norm for matrices |
| $\tilde{\mathcal{O}}$ | The Landau notation, ignoring logarithmic terms |
| $\phi$ | The approximation function for $\tan$, i.e., $\phi(\theta) = \frac{\theta}{1-\theta}$ |
| $\mathrm{Avg}_{\boldsymbol{x},\boldsymbol{x}'\in\mathcal{S}_k} h(\boldsymbol{x},\boldsymbol{x}')$ | The average over pairs of different elements |
| $[i]$ | The set from 1 to $i$, i.e., $\{1,\dots,i\}$ |

Table 4: Notation table for data

| | |
|---|---|
| $\mathcal{X}, \mathcal{Y}$ | An instance space and a label space |
| $\boldsymbol{x}, y$ | An instance and a label |
| $K$ | The number of classes, i.e., $|\mathcal{Y}|$ |
| $p$ | The instance dimension |
| $P, P_k$ | A sample distribution and a class-conditional distribution |
| $\mathcal{S}, \mathcal{S}_k$ | A training dataset and a class-specific training dataset |
| $\tilde{\mathcal{S}}_k$ | A class-specific test dataset |
| $N, n_k$ | The size of $\mathcal{S}$ and $\mathcal{S}_k$ |
| $\tilde{n}_k$ | The size of $\tilde{\mathcal{S}}_k$ |
| $\rho$ | An imbalance factor, $\rho \equiv \frac{n_1}{n_K} = \frac{\max_k n_k}{\min_k n_k}$ |
| $\boldsymbol{\sigma}, \sigma_j$ | Rademacher random variables |

Table 5: Notation table for models

| | |
|---|---|
| $\boldsymbol{f}, \overline{\boldsymbol{f}}$ | A feature map and a ReLU neural feature map |
| $\|\boldsymbol{f}\|$ | The value of $\|\boldsymbol{f}(\boldsymbol{x})\|$, assuming that $\|\boldsymbol{f}(\boldsymbol{x})\|$ is equal for all $\boldsymbol{x} \in \mathcal{X}$ |
| $\mathcal{F}, \overline{\mathcal{F}}, \overline{\mathcal{F}}^M$ | The set of feature maps, ReLU neural feature maps, and ReLU neural feature maps with constraints on $M$ |
| $\boldsymbol{h}, \overline{\boldsymbol{h}}$ | The output of the second-to-last layer of the feature map, and that of the ReLU neural feature map |
| $\boldsymbol{g}, g_k$ | A logit and the logit for class $k$ |
| $\boldsymbol{W}, \boldsymbol{w}_k$ | The weights of a linear classifier and its vector for class $k$ |
| $\boldsymbol{W}^s$ | The $s$-th linear layer from the end of the feature map |
| $r$ | The rank of $\boldsymbol{W}^1$ |
| $s_l, \boldsymbol{u}_l, \boldsymbol{v}_l$ | The $l$-th singular value, left singular vector, and right singular vector of $\boldsymbol{W}^1$, from its singular value decomposition |
| $\mathcal{H}$ | The set of $\boldsymbol{h}$ |
| $\mathcal{H}_l, \overline{\mathcal{H}}_l^M$ | The set of $(\boldsymbol{v}_l, \boldsymbol{h})$ for $\boldsymbol{f} \in \mathcal{F}$ and set of $(\boldsymbol{v}_l, \overline{\boldsymbol{h}})$ for $\overline{\boldsymbol{f}} \in \overline{\mathcal{F}}^M$ |
| $\mathfrak{R}_{n_k}(\mathcal{H}_l), \hat{\mathfrak{R}}_{\mathcal{S}_k}(\mathcal{H}_l)$ | The Rademacher complexity and the empirical Rademacher complexity for $\mathcal{H}_l$ (see Section 4.1) |
| $\mathcal{C}(\mathcal{H}_l, \mathcal{X})$ | The complexity of $\mathcal{H}_l$ and $\mathcal{X}$ |
| $\bar{\mathcal{C}}(\mathcal{F}, \mathcal{X})$ | The mean of $\mathcal{C}(\mathcal{H}_l, \mathcal{X})$ for $l \in [r]$ |
| $q$ | The depth of the feature map |
| $d, d_1$ | The dimension of $\boldsymbol{f}(\boldsymbol{x})$ and the dimension of $\boldsymbol{h}(\boldsymbol{x})$ |
| $B$ | The upper bound of $\boldsymbol{h}$, i.e., $\sup_{\boldsymbol{x} \in \mathcal{X}, \boldsymbol{h} \in \mathcal{H}} \|\boldsymbol{h}(\boldsymbol{x})\| \le B$ |
| $M$ | The upper bound of $\prod_{s=2}^q \|\boldsymbol{W}^s\|_F$ |
| $\boldsymbol{\mu}_{\boldsymbol{f}}(P), \boldsymbol{\mu}(P)$ | The expectation of $\boldsymbol{f}(\boldsymbol{x})$ for $\boldsymbol{x} \sim P$ |
| $\boldsymbol{\mu}_{\boldsymbol{f}}(\mathcal{S}), \boldsymbol{\mu}(\mathcal{S})$ | The mean of $\boldsymbol{f}(\boldsymbol{x})$ for $\boldsymbol{x} \in \mathcal{S}$ |
| $\overline{\boldsymbol{\mu}}(P), \overline{\boldsymbol{\mu}}(\mathcal{S})$ | Same as $\boldsymbol{\mu}_{\overline{\boldsymbol{f}}}(P)$ and $\boldsymbol{\mu}_{\overline{\boldsymbol{f}}}(\mathcal{S})$ |
| $\Pi(\theta; k), \overline{\Pi}(\theta; k)$ | The angular bound probability for $\boldsymbol{f}$ and for $\overline{\boldsymbol{f}}$ (see Section 4.2). |
| $\gamma_{1v1}, \gamma_{\times}, \gamma_{+}$ | Hyperparameters for 1vs1adjuster, MLA, and ALA |
| $\theta_{k,k'}, \theta_{k,k'}^*, \theta_{k,k'}^*(\gamma_{1v1})$ | The angle of the decision boundary, the optimal angle, and the optimal angle adjusted by $\gamma_{1v1}$ |
| $\theta_{k,k'}^{\times}, \theta_{k,k'}^{+}$ | The angles of the decision boundaries derived from MLA and ALA |
| $\Theta, \Theta^*, \Theta_{\gamma_{1v1}}^*$ | The set of $\theta_{k,k'}, \theta_{k,k'}^*$, and $\theta_{k,k'}^*(\gamma_{1v1})$ |
| $\psi$ | The angle between the weight vectors of an ETF classifier, i.e., $\psi = \arccos\left(-\frac{1}{K-1}\right)$ |
| $\tau_{k,k'}$ | The ratio $\frac{\|\boldsymbol{\mu}(\mathcal{S}_k)\| n_k^{\gamma_{1v1}}}{\|\boldsymbol{\mu}(\mathcal{S}_{k'})\| n_{k'}^{\gamma_{1v1}}}$ |
| $\boldsymbol{m}_{k,k'}$ | The normal vector of the decision boundary between $k$ and $k'$ |

# B   RELATED WORK

## B.1   MULTIPLICATIVE LOGIT ADJUSTMENT

MLA has been validated for its effectiveness across various studies. Hasegawa & Sato (2023) demonstrated that MLA performs on par with or better than ALA under realistic conditions. Additionally, Ye et al. (2022) highlighted significant differences in the feature space occupied by each class in LTR, confirming the effectiveness of class-dependent temperatures, which is equivalent to MLA. Beyond image classification, various studies suggest the utility of MLA-like adjustments in other fields. For instance, in object detection, Alexandridis et al. (2023) proposed an MLA variant based on object occurrence frequency, called inverse image frequency. Similarly, in text retrieval tasks, the inverse document frequency (Salton & Buckley, 1988), which adjusts the score function by multiplying the inverse frequency of words in a document, is widely used. Thus, MLA is not only applicable to LTR in classification problems but also serves as a fundamental technique with potential applications across a broad range of domains.

## C  PROOF

Here, we present the detailed proofs and supplementary explanations for the theories proposed in this paper. Appendix C.1 contains the proofs for the propositions discussed in Section 4.2. Appendix C.2 provides specific examples of applying our theory to ReLU neural networks. Appendix C.3 includes the proof of the lemma presented in Section 4.3.

### C.1  MAIN ANALYSIS

For all $k$ where $n_k > 1$, the operation of averaging a pair of elements $\boldsymbol{x}, \boldsymbol{x}' \in \mathcal{S}_k$ is defined as follows:

$$\mathrm{Avg}_{\boldsymbol{x},\boldsymbol{x}'\in\mathcal{S}_k}\, l(\boldsymbol{x},\boldsymbol{x}') \equiv \frac{1}{n_k(n_k-1)} \sum_{\substack{\boldsymbol{x}_{j_1}\in\mathcal{S}_k \\ \boldsymbol{x}_{j_2}\in\mathcal{S}_k\setminus\{\boldsymbol{x}_{j_1}\}}} l(\boldsymbol{x}_{j_1}, \boldsymbol{x}_{j_2}). \tag{23}$$

First, we propose two lemmas required for the proof of the propositions in Section 4.2. Then, we prove the propositions using the lemmas.

**Lemma 3.** *Suppose $n_k > 2$. For any $\boldsymbol{f} \in \mathcal{F}, \delta > 0$, the following holds with a probability of at least $1 - \delta$:*

$$\mathbb{E}_{\boldsymbol{x},\boldsymbol{x}'\sim P_k}[\|\boldsymbol{f}(\boldsymbol{x}) - \boldsymbol{f}(\boldsymbol{x}')\|] - \|\boldsymbol{W}^1\|_2 \sum_{l=1}^r \mathrm{Avg}_{\boldsymbol{x},\boldsymbol{x}'\in\mathcal{S}_k} |\boldsymbol{v}_l^\top(\boldsymbol{h}(\boldsymbol{x}) - \boldsymbol{h}(\boldsymbol{x}'))|$$

$$< \frac{r\|\boldsymbol{W}^1\|_2}{\sqrt{n_k}}\left(4\bar{\mathcal{C}}(\mathcal{F},\mathcal{X}) + 4B + B\sqrt{2\log\frac{r}{\delta}}\right). \tag{24}$$

*Proof.* We derive this lemma following the proof of Theorem 3.3 in Mohri et al. (2012). By the triangle inequality, the following holds:

$$\|\boldsymbol{f}(\boldsymbol{x}) - \boldsymbol{f}(\boldsymbol{x}')\| = \left\|\sum_{l=1}^r s_l \boldsymbol{u}_l \boldsymbol{v}_l^\top(\boldsymbol{h}(\boldsymbol{x}) - \boldsymbol{h}(\boldsymbol{x}'))\right\|$$

$$\leq \sum_{l=1}^r s_l \|\boldsymbol{u}_l\| |\boldsymbol{v}_l^\top(\boldsymbol{h}(\boldsymbol{x}) - \boldsymbol{h}(\boldsymbol{x}'))|$$

$$\leq \|\boldsymbol{W}^1\|_2 \sum_{l=1}^r |\boldsymbol{v}_l^\top(\boldsymbol{h}(\boldsymbol{x}) - \boldsymbol{h}(\boldsymbol{x}'))|. \tag{25}$$

We use this inequality to bound $\mathbb{E}_{\boldsymbol{x},\boldsymbol{x}'\sim P_k}[\|\boldsymbol{f}(\boldsymbol{x}) - \boldsymbol{f}(\boldsymbol{x}')\|]$. Then, for each $l \in [r]$, we calculate the probabilistic upper bound of $\mathbb{E}_{\boldsymbol{x},\boldsymbol{x}'\sim P_k}[|\boldsymbol{v}_l^\top(\boldsymbol{h}(\boldsymbol{x}) - \boldsymbol{h}(\boldsymbol{x}'))|] - \mathrm{Avg}_{\boldsymbol{x},\boldsymbol{x}'\in\mathcal{S}_k}|\boldsymbol{v}_l^\top(\boldsymbol{h}(\boldsymbol{x}) - \boldsymbol{h}(\boldsymbol{x}'))|$ and apply the union bound to complete the proof.

Note that $\mathbb{E}_{\boldsymbol{x},\boldsymbol{x}'\sim P_k}[|\boldsymbol{v}_l^\top(\boldsymbol{h}(\boldsymbol{x}) - \boldsymbol{h}(\boldsymbol{x}'))|] = \mathbb{E}_{\mathcal{S}_k\sim P_k^{n_k}}[\mathrm{Avg}_{\boldsymbol{x},\boldsymbol{x}'\in\mathcal{S}_k}|\boldsymbol{v}_l^\top(\boldsymbol{h}(\boldsymbol{x}) - \boldsymbol{h}(\boldsymbol{x}'))|]$. For $\mathcal{S}_k^1, \mathcal{S}_k^2 \sim P_k^{n_k}$, define $\Phi_l(\mathcal{S}_k^1)$ as follows:

$$\Phi_l(\mathcal{S}_k^1) = \sup_{(\boldsymbol{v}_l,\boldsymbol{h})\in\mathcal{H}_l}\left(\mathbb{E}_{\boldsymbol{x},\boldsymbol{x}'\sim P_k}[|\boldsymbol{v}_l^\top(\boldsymbol{h}(\boldsymbol{x}) - \boldsymbol{h}(\boldsymbol{x}'))|] - \mathrm{Avg}_{\boldsymbol{x},\boldsymbol{x}'\in\mathcal{S}_k^1}|\boldsymbol{v}_l^\top(\boldsymbol{h}(\boldsymbol{x}) - \boldsymbol{h}(\boldsymbol{x}'))|\right). \tag{26}$$

Let $\mathcal{S}_k^1, \mathcal{S}_k^2 \sim P_k^{n_k}$ be datasets that differ only in the $j$-th sample $\boldsymbol{x}_j^1, \boldsymbol{x}_j^2$, and define $\Phi_l(\mathcal{S}_k^2)$ similarly as $\Phi_l(\mathcal{S}_k^1)$. The difference between $\Phi_l(\mathcal{S}_k^1)$ and $\Phi_l(\mathcal{S}_k^2)$ can be upper-bounded as follows:

$$
\begin{aligned}
\Phi_l(\mathcal{S}_k^1) - \Phi_l(\mathcal{S}_k^2) \leq &\sup_{(\boldsymbol{v}_l, \boldsymbol{h}) \in \mathcal{H}_l} \Big( \mathrm{Avg}_{\boldsymbol{x}, \boldsymbol{x}' \in \mathcal{S}_k^1} \big| \boldsymbol{v}_l^\top (\boldsymbol{h}(\boldsymbol{x}) - \boldsymbol{h}(\boldsymbol{x}')) \big| \\
&\qquad - \mathrm{Avg}_{\boldsymbol{x}, \boldsymbol{x}' \in \mathcal{S}_k^2} \big| \boldsymbol{v}_l^\top (\boldsymbol{h}(\boldsymbol{x}) - \boldsymbol{h}(\boldsymbol{x}')) \big| \Big) \\
= &\frac{1}{n_k(n_k - 1)} \sum_{i \neq j} \sup_{(\boldsymbol{v}_l, \boldsymbol{h}) \in \mathcal{H}_l} \big( |\boldsymbol{v}_l^\top (\boldsymbol{h}(\boldsymbol{x}_j^1) - \boldsymbol{h}(\boldsymbol{x}_i))| - |\boldsymbol{v}_l^\top (\boldsymbol{h}(\boldsymbol{x}_j^2) - \boldsymbol{h}(\boldsymbol{x}_i))| \big) \\
\leq &\frac{1}{n_k(n_k - 1)} \sum_{i \neq j} \sup_{(\boldsymbol{v}_l, \boldsymbol{h}) \in \mathcal{H}_l} \big( |\boldsymbol{v}_l^\top (\boldsymbol{h}(\boldsymbol{x}_j^1) - \boldsymbol{h}(\boldsymbol{x}_j^2))| \big) \\
\leq &\frac{2}{n_k} \sup_{\boldsymbol{x} \in \mathcal{X}, (\boldsymbol{v}_l, \boldsymbol{h}) \in \mathcal{H}_l} \| \boldsymbol{h}(\boldsymbol{x}) \| \qquad\qquad\qquad (27) \\
\leq &\frac{2B}{n_k}. \qquad\qquad\qquad\qquad\qquad\qquad\qquad\qquad (28)
\end{aligned}
$$

Therefore, using McDiarmid's inequality, we can prove that for any $\delta > 0$ the following holds with probability at least $1 - \delta$:

$$
\Phi_l(\mathcal{S}_k) \leq \mathbb{E}_{\mathcal{S}_k}[\Phi_l(\mathcal{S}_k)] + B\sqrt{\frac{2 \log \frac{1}{\delta}}{n_k}}. \qquad (29)
$$

Note that for any $\boldsymbol{f} \in \mathcal{F}$, the following inequality holds:

$$
\mathbb{E}_{\boldsymbol{x}, \boldsymbol{x}' \sim P_k} \big[ |\boldsymbol{v}_l^\top (\boldsymbol{h}(\boldsymbol{x}) - \boldsymbol{h}(\boldsymbol{x}'))| \big] - \mathrm{Avg}_{\boldsymbol{x}, \boldsymbol{x}' \in \mathcal{S}_k^1} \big| \boldsymbol{v}_l^\top (\boldsymbol{h}(\boldsymbol{x}) - \boldsymbol{h}(\boldsymbol{x}')) \big| \leq \Phi_l(\mathcal{S}_k).
$$

Using Eq. (25), we can derive the following:

$$
\text{Left hand side of Eq. (24)} \leq \| \boldsymbol{W}^1 \|_2 \sum_{l=1}^r \Phi_l(\mathcal{S}_k). \qquad (30)
$$

Thus, we aim to take a union bound of Eq. (29) over all $l$ to derive an upper bound.

Next, we bound $\mathbb{E}_{\mathcal{S}_k}[\Phi_l(\mathcal{S}_k)]$:

$$
\begin{aligned}
\mathbb{E}_{\mathcal{S}_k}[\Phi_l(\mathcal{S}_k)] = \mathbb{E}_{\mathcal{S}_k} \Bigg[ &\sup_{(\boldsymbol{v}_l, \boldsymbol{h}) \in \mathcal{H}_l} \Big( \mathbb{E}_{\boldsymbol{x}, \boldsymbol{x}' \sim P_k} \big[ |\boldsymbol{v}_l^\top (\boldsymbol{h}(\boldsymbol{x}) - \boldsymbol{h}(\boldsymbol{x}'))| \big] \\
&\qquad - \mathrm{Avg}_{\boldsymbol{x}, \boldsymbol{x}' \in \mathcal{S}_k} \big| \boldsymbol{v}_l^\top (\boldsymbol{h}(\boldsymbol{x}) - \boldsymbol{h}(\boldsymbol{x}')) \big| \Big) \Bigg] \\
\leq \mathbb{E}_{\mathcal{S}_k, \hat{\mathcal{S}}_k} \Bigg[ &\sup_{(\boldsymbol{v}_l, \boldsymbol{h}) \in \mathcal{H}_l} \Big( \mathrm{Avg}_{\boldsymbol{x}, \boldsymbol{x}' \in \hat{\mathcal{S}}_k} \big| \boldsymbol{v}_l^\top (\boldsymbol{h}(\boldsymbol{x}) - \boldsymbol{h}(\boldsymbol{x}')) \big| \\
&\qquad - \mathrm{Avg}_{\boldsymbol{x}, \boldsymbol{x}' \in \mathcal{S}_k} \big| \boldsymbol{v}_l^\top (\boldsymbol{h}(\boldsymbol{x}) - \boldsymbol{h}(\boldsymbol{x}')) \big| \Big) \Bigg] \\
= \mathbb{E}_{\mathcal{S}_k, \hat{\mathcal{S}}_k} \Bigg[ &\sup_{(\boldsymbol{v}_l, \boldsymbol{h}) \in \mathcal{H}_l} \frac{1}{n_k(n_k - 1)} \Bigg( \Bigg( \sum_{\substack{\hat{\boldsymbol{x}}_j \in \hat{\mathcal{S}}_k \\ \hat{\boldsymbol{x}}_{j'} \in \hat{\mathcal{S}}_k \setminus \{\hat{\boldsymbol{x}}_j\}}} |\boldsymbol{v}_l^\top (\boldsymbol{h}(\hat{\boldsymbol{x}}_j) - \boldsymbol{h}(\hat{\boldsymbol{x}}_{j'}))| \Bigg) \\
&\qquad\qquad - \Bigg( \sum_{\substack{\boldsymbol{x}_j \in \mathcal{S}_k \\ \boldsymbol{x}_{j'} \in \mathcal{S}_k \setminus \{\boldsymbol{x}_j\}}} |\boldsymbol{v}_l^\top (\boldsymbol{h}(\boldsymbol{x}_j) - \boldsymbol{h}(\boldsymbol{x}_{j'}))| \Bigg) \Bigg) \Bigg]
\end{aligned}
$$

$$
= \mathbb{E}_{\boldsymbol{\sigma}, \mathcal{S}_k, \hat{\mathcal{S}}_k} \left[ \sup_{(\boldsymbol{v}_l, \boldsymbol{h}) \in \mathcal{H}_l} \frac{1}{n_k(n_k - 1)} \left( \left( \sum_{\substack{\hat{\boldsymbol{x}}_j \in \hat{\mathcal{S}}_k \\ \hat{\boldsymbol{x}}_{j'} \in \hat{\mathcal{S}}_k \setminus \{\hat{\boldsymbol{x}}_j\}}} \sigma_j |\boldsymbol{v}_l^\top (\boldsymbol{h}(\hat{\boldsymbol{x}}_j) - \boldsymbol{h}(\hat{\boldsymbol{x}}_{j'}))| \right) \right. \right.
$$

$$
\left. \left. - \left( \sum_{\substack{\boldsymbol{x}_j \in \mathcal{S}_k \\ \boldsymbol{x}_{j'} \in \mathcal{S}_k \setminus \{\boldsymbol{x}_j\}}} \sigma_j |\boldsymbol{v}_l^\top (\boldsymbol{h}(\boldsymbol{x}_j) - \boldsymbol{h}(\boldsymbol{x}_{j'}))| \right) \right) \right]
$$

$$
\leq 2 \mathbb{E}_{\boldsymbol{\sigma}, \mathcal{S}_k} \left[ \sup_{(\boldsymbol{v}_l, \boldsymbol{h}) \in \mathcal{H}_l} \frac{1}{n_k(n_k - 1)} \sum_{\substack{\boldsymbol{x}_j \in \mathcal{S}_k \\ \boldsymbol{x}_{j'} \in \mathcal{S}_k \setminus \{\hat{\boldsymbol{x}}_j\}}} \sigma_j |\boldsymbol{v}_l^\top (\boldsymbol{h}(\boldsymbol{x}_j) - \boldsymbol{h}(\boldsymbol{x}_{j'}))| \right]
$$

$$
= 2 \mathbb{E}_{\boldsymbol{\sigma}, \mathcal{S}_k} \left[ \sup_{(\boldsymbol{v}_l, \boldsymbol{h}) \in \mathcal{H}_l} \frac{1}{n_k(n_k - 1)} \sum_{\substack{\boldsymbol{x}_j \in \mathcal{S}_k \\ \boldsymbol{x}_{j'} \in \mathcal{S}_k}} \sigma_j |\boldsymbol{v}_l^\top (\boldsymbol{h}(\boldsymbol{x}_j) - \boldsymbol{h}(\boldsymbol{x}_{j'}))| \right]
$$

$$
\leq \frac{2}{n_k - 1} \mathbb{E}_{\mathcal{S}_k} \left[ \sum_{\boldsymbol{x}_{j'} \in \mathcal{S}_k} \mathbb{E}_{\boldsymbol{\sigma}} \left[ \sup_{(\boldsymbol{v}_l, \boldsymbol{h}) \in \mathcal{H}_l} \frac{1}{n_k} \sum_{\boldsymbol{x}_j \in \mathcal{S}_k} \sigma_j |\boldsymbol{v}_l^\top (\boldsymbol{h}(\boldsymbol{x}_j) - \boldsymbol{h}(\boldsymbol{x}_{j'}))| \right] \right]
$$

$$
\leq \frac{2}{n_k - 1} \mathbb{E}_{\mathcal{S}_k} \left[ \sum_{\boldsymbol{x}_{j'} \in \mathcal{S}_k} \mathbb{E}_{\boldsymbol{\sigma}} \left[ \sup_{(\boldsymbol{v}_l, \boldsymbol{h}) \in \mathcal{H}_l} \frac{1}{n_k} \sum_{\boldsymbol{x}_j \in \mathcal{S}_k} \sigma_j \boldsymbol{v}_l^\top (\boldsymbol{h}(\boldsymbol{x}_j) - \boldsymbol{h}(\boldsymbol{x}_{j'})) \right] \right] \tag{31}
$$

$$
\leq \frac{2 n_k}{n_k - 1} \mathbb{E}_{\boldsymbol{\sigma}, \mathcal{S}_k} \left[ \sup_{(\boldsymbol{v}_l, \boldsymbol{h}) \in \mathcal{H}_l} \frac{1}{n_k} \sum_{\boldsymbol{x}_j \in \mathcal{S}_k} \sigma_j \boldsymbol{v}_l^\top \boldsymbol{h}(\boldsymbol{x}_j) \right]
$$

$$
+ \frac{2}{n_k(n_k - 1)} \mathbb{E}_{\mathcal{S}_k} \left[ \sum_{\boldsymbol{x}_{j'} \in \mathcal{S}_k} \sup_{(\boldsymbol{v}_l, \boldsymbol{h}) \in \mathcal{H}_l} |\boldsymbol{v}_l^\top \boldsymbol{h}(\boldsymbol{x}_{j'})| \right] \mathbb{E}_{\boldsymbol{\sigma}} \left[ \left| \sum_{j=1}^{n_k} -\sigma_j \right| \right]
$$

$$
\leq \frac{2}{n_k - 1} \left( n_k \mathfrak{R}_{n_k}(\mathcal{H}_l) + B \mathbb{E}_{\boldsymbol{\sigma}} \left[ \left| \sum_{j=1}^{n_k} \sigma_j \right| \right] \right). \tag{32}
$$

The transformation in Eq. (31) is similar to that of Talagrand's contraction lemma (Ledoux & Talagrand, 1991).

In addition, the following inequality holds:

$$
\mathbb{E}_{\boldsymbol{\sigma}} \left[ \left| \sum_{j=1}^{n_k} \sigma_j \right| \right] = \sqrt{\mathbb{E}_{\boldsymbol{\sigma}} \left[ \left| \sum_{j=1}^{n_k} \sigma_j \right| \right]^2}
$$

$$
\leq \sqrt{\mathbb{E}_{\boldsymbol{\sigma}} \left[ \left| \sum_{j=1}^{n_k} \sigma_j \right|^2 \right]}
$$

$$
= \sqrt{\mathbb{E}_{\boldsymbol{\sigma}} \left[ \sum_{j=1}^{n_k} \sigma_j^2 + 2 \sum_{j \neq j'} \sigma_j \sigma_{j'} \right]}
$$

$$
= \sqrt{n_k}. \tag{33}
$$

From Eqs. (29), (32), and (33), it is proven that following holds with probability at least $1 - \delta$,

$$
\begin{aligned}
\Phi_l(\mathcal{S}_k) &\leq \frac{2n_k}{n_k - 1}\left(\mathfrak{R}_{n_k}(\mathcal{H}_l) + \frac{B}{\sqrt{n_k}}\right) + \frac{B}{\sqrt{n_k}}\sqrt{2\log\frac{1}{\delta}} \\
&\leq \frac{2n_k}{(n_k - 1)\sqrt{n_k}}(\mathcal{C}(\mathcal{H}_l, \mathcal{X}) + B) + \frac{B}{\sqrt{n_k}}\sqrt{2\log\frac{1}{\delta}} \\
&< \frac{1}{\sqrt{n_k}}\left(4\mathcal{C}(\mathcal{H}_l, \mathcal{X}) + 4B + B\sqrt{2\log\frac{1}{\delta}}\right).
\end{aligned}
\tag{34}
$$

Substituting Eq. (34) into Eq. (30) and taking the union bound for all $l$ proves this lemma.

$\square$

**Lemma 4.** *Suppose $n_k > 2$. For $\boldsymbol{f} \in \mathcal{F}$, assume that for all $\boldsymbol{x} \in \mathcal{S}_k$, $\boldsymbol{f}(\boldsymbol{x}) = \boldsymbol{\mu}(\mathcal{S}_k)$ holds. For any $0 < \delta, \delta' < 1$, define $\theta$ as follows:*

$$
\theta \equiv \frac{\pi}{2\delta'}\frac{r\|\boldsymbol{W}^1\|_2}{\sqrt{n_k}\|\boldsymbol{\mu}(\mathcal{S}_k)\|}\left(4\bar{\mathcal{C}}(\mathcal{F}, \mathcal{X}) + 4B + B\sqrt{2\log\frac{r}{\delta}}\right).
\tag{35}
$$

*Assume that $\delta, \delta'$ satisfy $\theta \leq \frac{\pi}{2}$. For $\boldsymbol{x} \sim P_k$, the probability that the angle between $\boldsymbol{f}(\boldsymbol{x})$ and $\mu(\mathcal{S}_k)$ is less than $\theta$ is at least $1 - (\delta + \delta')$.*

*Proof.* We bound $\mathbb{E}_{\boldsymbol{x} \sim P_k, \mathcal{S}_k \sim P_k^{n_k}}[\|\boldsymbol{f}(\boldsymbol{x}) - \boldsymbol{\mu}(\mathcal{S}_k)\|]$ and apply Markov's inequality. Expanding this,

$$
\begin{aligned}
\mathbb{E}_{\boldsymbol{x} \sim P_k, \mathcal{S}_k \sim P_k^{n_k}}[\|\boldsymbol{f}(\boldsymbol{x}) - \boldsymbol{\mu}(\mathcal{S}_k)\|] &= \frac{1}{n_k}\mathbb{E}_{\boldsymbol{x} \sim P_k, \mathcal{S}_k \sim P_k^{n_k}}\left[\left\|n_k\boldsymbol{f}(\boldsymbol{x}) - \sum_{\boldsymbol{x}' \in \mathcal{S}_k}\boldsymbol{f}(\boldsymbol{x}')\right\|\right] \\
&\leq \frac{1}{n_k}\mathbb{E}_{\boldsymbol{x} \sim P_k, \mathcal{S}_k \sim P_k^{n_k}}\left[\sum_{\boldsymbol{x}' \in \mathcal{S}_k}\|\boldsymbol{f}(\boldsymbol{x}) - \boldsymbol{f}(\boldsymbol{x}')\|\right] \\
&= \mathbb{E}_{\boldsymbol{x}, \boldsymbol{x}' \sim P_k}[\|\boldsymbol{f}(\boldsymbol{x}) - \boldsymbol{f}(\boldsymbol{x}')\|].
\end{aligned}
\tag{36}
$$

Using Markov's inequality, for any $\delta' > 0$, with probability at least $1 - \delta'$,

$$
\begin{aligned}
\|\boldsymbol{f}(\boldsymbol{x}) - \boldsymbol{\mu}(\mathcal{S}_k)\| &\leq \frac{1}{\delta'}\mathbb{E}_{\boldsymbol{x} \sim P_k, \mathcal{S}_k \sim P_k^{n_k}}[\|\boldsymbol{f}(\boldsymbol{x}) - \boldsymbol{\mu}(\mathcal{S}_k)\|] \\
&= \frac{1}{\delta'}\mathbb{E}_{\boldsymbol{x}, \boldsymbol{x}' \sim P_k}[\|\boldsymbol{f}(\boldsymbol{x}) - \boldsymbol{f}(\boldsymbol{x}')\|].
\end{aligned}
\tag{37}
$$

Before applying Lemma 3 to this case, we demonstrate that the second term on the left-hand side of Eq. (24) is zero in this situation. When $\boldsymbol{f}(\boldsymbol{x}) = \boldsymbol{\mu}(\mathcal{S}_k)$ satisfies for all $\boldsymbol{x} \in \mathcal{S}_k$, $\boldsymbol{f}(\boldsymbol{x}) = \boldsymbol{f}(\boldsymbol{x}')$ holds for all $\boldsymbol{x}, \boldsymbol{x}' \in \mathcal{S}_k$. Then, we have $\sum_{l=1}^r s_l \boldsymbol{u}_l \boldsymbol{v}_l^\top(\boldsymbol{h}(\boldsymbol{x}) - \boldsymbol{h}(\boldsymbol{x}')) = \boldsymbol{0}$. Since $\boldsymbol{u}_l$ are the left singular vectors of $\boldsymbol{W}^1$, they are linearly independent for different $l$. Therefore, for all $l \in [r]$ and for all $\boldsymbol{x}, \boldsymbol{x}' \in \mathcal{S}_k$,

$$
\boldsymbol{v}_l^\top(\boldsymbol{h}(\boldsymbol{x}) - \boldsymbol{h}(\boldsymbol{x}')) = 0.
\tag{38}
$$

Then, we combine the equation Eqs. (37) and (38) with the result of Lemma 3 by union bound. For any $\delta, \delta' > 0$, with probability at least $1 - (\delta + \delta')$, the following holds:

$$
\begin{aligned}
\|\boldsymbol{f}(\boldsymbol{x}) - \boldsymbol{\mu}(\mathcal{S}_k)\| &< \frac{1}{\delta'}\frac{r\|\boldsymbol{W}^1\|_2}{\sqrt{n_k}}\left(4\bar{\mathcal{C}}(\mathcal{F}, \mathcal{X}) + 4B + B\sqrt{2\log\frac{r}{\delta}}\right) \\
&\equiv \frac{2}{\pi}\|\boldsymbol{\mu}(\mathcal{S}_k)\|\theta.
\end{aligned}
\tag{39}
$$

Since $1 - (\delta + \delta')$ is a lower bound for the probability, even restricting $\delta, \delta' < 1$ does not compromise the usefulness of the theorem.

The angle between $\boldsymbol{f}(\boldsymbol{x})$ and $\boldsymbol{\mu}(\mathcal{S}_k)$ can be expressed as $\arcsin\left(\frac{\|\boldsymbol{f}(\boldsymbol{x})-\boldsymbol{\mu}(\mathcal{S}_k)\|}{\|\boldsymbol{\mu}(\mathcal{S}_k)\|}\right)$ when $\frac{\|\boldsymbol{f}(\boldsymbol{x})-\boldsymbol{\mu}(\mathcal{S}_k)\|}{\|\boldsymbol{\mu}(\mathcal{S}_k)\|} \leq 1$. Since $\arcsin(t)$ is strictly monotonically increasing for $0 \leq t \leq 1$, when we have $\theta \leq \frac{\pi}{2}$, the upper bound of the angle between $\boldsymbol{f}(\boldsymbol{x})$ and $\boldsymbol{\mu}(\mathcal{S}_k)$ can be written as $\arcsin\left(\frac{\|\boldsymbol{f}(\boldsymbol{x})-\boldsymbol{\mu}(\mathcal{S}_k)\|}{\|\boldsymbol{\mu}(\mathcal{S}_k)\|}\right) < \arcsin\left(\frac{2}{\pi}\theta\right) \leq \theta$. This means that the upper bound of the angle between the features and the average training features of the class is less than $\theta$ with probability at least $1 - (\delta + \delta')$.

$\square$

**Proposition 1.** *Suppose $n_k > 2$. For $\boldsymbol{f} \in \mathcal{F}$, assume that for all $\boldsymbol{x} \in \mathcal{S}_k$, $\boldsymbol{f}(\boldsymbol{x}) = \boldsymbol{\mu}(\mathcal{S}_k)$ holds. Consider any $\theta$ that satisfies the following condition:*

$$\frac{\pi}{2} \frac{r\|\boldsymbol{W}^1\|_2}{\sqrt{n_k}\|\boldsymbol{\mu}(\mathcal{S}_k)\|}\left(4\bar{\mathcal{C}}(\mathcal{F},\mathcal{X}) + 4B + B\sqrt{2\log\frac{\sqrt{n_k}\|\boldsymbol{\mu}(\mathcal{S}_k)\|}{\|\boldsymbol{W}^1\|_2}}\right) < \theta < \frac{\pi}{2}, \tag{5}$$

*where $\|\cdot\|_2$ denotes the spectral norm. For such $\theta$, the following holds:*

$$\Pi(\theta;k) = 1 - \frac{\pi}{2\theta}\frac{r\|\boldsymbol{W}^1\|_2}{\sqrt{n_k}\|\boldsymbol{\mu}(\mathcal{S}_k)\|}\left(4\bar{\mathcal{C}}(\mathcal{F},\mathcal{X}) + 4B + 1 + B\sqrt{2\log\frac{2\theta\sqrt{n_k}\|\boldsymbol{\mu}(\mathcal{S}_k)\|}{\pi\|\boldsymbol{W}^1\|_2}}\right). \tag{6}$$

*Proof.* Consider the following $\delta, \delta' > 0$:

$$\delta = \frac{\pi}{2\theta}\frac{r\|\boldsymbol{W}^1\|_2}{\sqrt{n_k}\|\boldsymbol{\mu}(\mathcal{S}_k)\|}, \tag{40}$$

$$\delta' = \frac{\pi}{2\theta}\frac{r\|\boldsymbol{W}^1\|_2}{\sqrt{n_k}\|\boldsymbol{\mu}(\mathcal{S}_k)\|}\left(4\bar{\mathcal{C}}(\mathcal{F},\mathcal{X}) + 4B + B\sqrt{2\log\frac{2\theta\sqrt{n_k}\|\boldsymbol{\mu}(\mathcal{S}_k)\|}{\pi\|\boldsymbol{W}^1\|_2}}\right). \tag{41}$$

Given $B \geq 1$ and the condition on $\theta$,

$$\delta < \delta' < \frac{\pi}{2\theta}\frac{r\|\boldsymbol{W}^1\|_2}{\sqrt{n_k}\|\boldsymbol{\mu}(\mathcal{S}_k)\|}\left(4\bar{\mathcal{C}}(\mathcal{F},\mathcal{X}) + 4B + B\sqrt{2\log\frac{\sqrt{n_k}\|\boldsymbol{\mu}(\mathcal{S}_k)\|}{\|\boldsymbol{W}^1\|_2}}\right) < 1. \tag{42}$$

Defining $\theta$ as above, we get $\theta = \frac{\pi}{2\delta'}\frac{r\|\boldsymbol{W}^1\|_2}{\sqrt{n_k}\|\boldsymbol{\mu}(\mathcal{S}_k)\|}\left(4\bar{\mathcal{C}}(\mathcal{F},\mathcal{X}) + 4B + B\sqrt{2\log\frac{r}{\delta}}\right)$. Therefore, by Lemma 4, the probability that the angle between $\boldsymbol{f}(\boldsymbol{x})$ and $\boldsymbol{\mu}(\mathcal{S}_k)$ is less than $\theta$ is at least

$$1 - (\delta + \delta') = 1 - \frac{\pi}{2\theta}\frac{r\|\boldsymbol{W}^1\|_2}{\sqrt{n_k}\|\boldsymbol{\mu}(\mathcal{S}_k)\|}\left(4\bar{\mathcal{C}}(\mathcal{F},\mathcal{X}) + 4B + 1 + B\sqrt{2\log\frac{2\theta\sqrt{n_k}\|\boldsymbol{\mu}(\mathcal{S}_k)\|}{\pi\|\boldsymbol{W}^1\|_2}}\right). \tag{43}$$

$\square$

**Proposition 2.** *Suppose $n_k, n_{k'} > 2$. For $\boldsymbol{f} \in \mathcal{F}$, assume that for both $\hat{k} \in \{k, k'\}$, $\boldsymbol{f}(\boldsymbol{x}) = \boldsymbol{\mu}(\mathcal{S}_{\hat{k}})$ holds for all $\boldsymbol{x} \in \mathcal{S}_{\hat{k}}$. Consider the following maximization problem:*

$$\max_{\theta_{k,k'},\theta_{k',k}} \Pi(\theta_{k,k'};k) + \Pi(\theta_{k',k};k') \qquad s.t.\ \theta_{k,k'}, \theta_{k',k} > 0, \quad \theta_{k,k'} + \theta_{k',k} = \psi. \tag{7}$$

*The unique solution $\theta^*_{k,k'}, \theta^*_{k',k}$ within the range where $\theta_{k,k'}$ and $\theta_{k',k}$ satisfy Eq. (5) is given by:*

$$\theta^*_{k,k'} = \psi\frac{\|\boldsymbol{\mu}(\mathcal{S}_{k'})\|\sqrt{n_{k'}}}{\|\boldsymbol{\mu}(\mathcal{S}_k)\|\sqrt{n_k} + \|\boldsymbol{\mu}(\mathcal{S}_{k'})\|\sqrt{n_{k'}}}, \quad \theta^*_{k',k} = \psi\frac{\|\boldsymbol{\mu}(\mathcal{S}_k)\|\sqrt{n_k}}{\|\boldsymbol{\mu}(\mathcal{S}_k)\|\sqrt{n_k} + \|\boldsymbol{\mu}(\mathcal{S}_{k'})\|\sqrt{n_{k'}}}. \tag{8}$$

*Proof.* Noting that $\theta_{k',k} = \psi - \theta_{k,k'}$, we have:

$$\frac{\partial}{\partial\theta_{k,k'}}(\Pi(\theta_{k,k'};k) + \Pi(\theta_{k',k};k')) = \left.\frac{\partial\Pi(\theta;k)}{\partial\theta}\right|_{\theta=\theta_{k,k'}} - \left.\frac{\partial\Pi(\theta;k')}{\partial\theta}\right|_{\theta=\psi-\theta_{k',k}}. \tag{44}$$

Since $\theta_{k,k'}^*, \theta_{k',k}^* > 0$ and $\theta_{k,k'}^* + \theta_{k',k}^* = \psi$, with $\theta_{k,k'}^* \|\boldsymbol{\mu}(\mathcal{S}_k)\|\sqrt{n_k} = \theta_{k',k}^* \|\boldsymbol{\mu}(\mathcal{S}_{k'})\|\sqrt{n_{k'}}$, the following holds:

$$\frac{\partial \Pi(\theta; k)}{\partial \theta}\bigg|_{\theta=\theta_{k,k'}^*} = \frac{\partial \Pi(\theta; k')}{\partial \theta}\bigg|_{\theta=\psi-\theta_{k',k}^*}. \tag{45}$$

Thus, we have $\frac{\partial}{\partial \theta_{k,k'}}(\Pi(\theta_{k,k'}; k) + \Pi(\theta_{k',k}; k')) = 0$ when $\theta_{k,k'} = \theta_{k,k'}^*$ and $\theta_{k',k} = \theta_{k',k}^*$.

Next, we prove that this is the only solution that gives the maximum value within the aforementioned range. We have:

$$\frac{\partial^2}{\partial \theta_{k,k'}^2}(\Pi(\theta_{k,k'}; y) + \Pi(\theta_{y'}; y')) = \frac{\partial^2 \Pi(\theta_{k,k'}; y)}{\partial \theta_{k,k'}^2} + \frac{\partial^2 \Pi(\theta; y')}{\partial \theta^2}\bigg|_{\theta=\psi-\theta_{k,k'}}. \tag{46}$$

Here, we define $\alpha_k \equiv \frac{\pi}{2}\frac{r\|\boldsymbol{W}^1\|_2}{\sqrt{n_k}\|\boldsymbol{\mu}(\mathcal{S}_k)\|}, \beta \equiv 4\bar{C}(\mathcal{F}, \mathcal{X}) + 4B + 1$, and $\zeta_k(\theta_{k,k'}) = 2\log\left(\frac{r\theta_{k,k'}}{\alpha_k}\right)$. Note that $\frac{\partial \zeta_k(\theta_{k,k'})}{\partial \theta_{k,k'}} = \frac{2}{\theta_{k,k'}}$. Then, the following holds:

$$\begin{aligned}
\frac{\partial^2 \Pi(\theta_{k,k'}; k)}{\partial \theta_{k,k'}^2} &= \frac{\partial^2}{\partial \theta_{k,k'}^2}\left(1 - \frac{\alpha_k}{\theta_{k,k'}}\left(\beta + B\sqrt{\zeta_k(\theta_{k,k'})}\right)\right) \\
&= \frac{\partial}{\partial \theta_{k,k'}}\left(\frac{\alpha_k}{\theta_{k,k'}^2}\left(\beta + B\zeta_k(\theta_{k,k'})^{\frac{1}{2}}\right) - \frac{B\alpha_k}{\theta_{k,k'}^2}\zeta_k(\theta_{k,k'})^{-\frac{1}{2}}\right) \\
&= \frac{\partial}{\partial \theta_{k,k'}}\left(\frac{\alpha_k}{\theta_{k,k'}^2}\left(\beta + B\zeta_k(\theta_{k,k'})^{\frac{1}{2}} - B\zeta_k(\theta_{k,k'})^{-\frac{1}{2}}\right)\right) \\
&= -\frac{2\alpha_k}{\theta_{k,k'}^3}\left(\beta + B\zeta_k(\theta_{k,k'})^{\frac{1}{2}} - B\zeta_k(\theta_{k,k'})^{-\frac{1}{2}}\right) \\
&\quad + \frac{\alpha_k}{\theta_{k,k'}^2}\left(\frac{B}{\theta_{k,k'}}\zeta_k(\theta_{k,k'})^{-\frac{1}{2}} + \frac{B}{\theta_{k,k'}}\zeta_k(\theta_{k,k'})^{-\frac{3}{2}}\right) \\
&= -\frac{\alpha_k}{\theta_{k,k'}^3}\left(2\beta + 2B\zeta_k(\theta_{k,k'})^{\frac{1}{2}} - 3B\zeta_k(\theta_{k,k'})^{-\frac{1}{2}} - B\zeta_k(\theta_{k,k'})^{-\frac{3}{2}}\right). \tag{47}
\end{aligned}$$

From the condition, there exists $\delta < 1$ such that $\theta_{k,k'} = \frac{\alpha_k}{\delta}$, implying $\frac{\theta_{k,k'}}{\alpha_k} > 1$. Since $\zeta_k(\theta_{k,k'}) > 2\log 2 > 2$, we have

$$\begin{aligned}
\frac{\partial^2 \Pi(\theta_{k,k'}; k)}{\partial \theta_{k,k'}^2} &\leq -\frac{\alpha_k}{\theta_{k,k'}^3}\left(2\beta + 2B\zeta_k(\theta_{k,k'})^{\frac{1}{2}} - 4B\zeta_k(\theta_{k,k'})^{-\frac{1}{2}}\right) \\
&= -\frac{2\alpha_k}{\theta_{k,k'}^3}\left(\beta + B\zeta_k(\theta_{k,k'})^{-\frac{1}{2}}(\zeta_k(\theta_{k,k'}) - 2)\right) \\
&< 0. \tag{48}
\end{aligned}$$

Similarly, we can prove that $\frac{\partial^2 \Pi(\theta; k')}{\partial \theta^2}\bigg|_{\theta=\psi-\theta_{k,k'}} < 0$.

Therefore, Eq. (8) is the only solution that gives the maximum value within the range where $\theta_{k,k'}$ and $\theta_{k',k}$ satisfy Eq. (5). $\qquad\square$

**Proposition 3.** *Suppose $n_k > 2$ for all $k \in \mathcal{Y}$. For $\boldsymbol{f} \in \mathcal{F}$, assume that for all $k \in \mathcal{Y}$, $\boldsymbol{f}(\boldsymbol{x}) = \boldsymbol{\mu}(\mathcal{S}_k)$ holds for all $\boldsymbol{x} \in \mathcal{S}_k$. Consider the following maximization problem:*

$$\max_{\Theta}\sum_{k\in\mathcal{Y}}\sum_{k'\neq k}\Pi(\theta_{k,k'}; k) + \Pi(\theta_{k',k}; k') \quad s.t. \ \forall k, k' \in \mathcal{Y}, \theta_{k,k'}, \theta_{k',k} > 0 \ and \ \theta_{k,k'} + \theta_{k',k} = \psi. \tag{9}$$

*The unique solution $\theta_{k,k'}^* \in \Theta^*$ within the range where all $\theta_{k,k'}^* \in \Theta^*$ satisfies Eq. (5) is given by:*

$$\theta_{k,k'}^* = \psi\frac{\|\boldsymbol{\mu}(\mathcal{S}_{k'})\|\sqrt{n_{k'}}}{\|\boldsymbol{\mu}(\mathcal{S}_k)\|\sqrt{n_k} + \|\boldsymbol{\mu}(\mathcal{S}_{k'})\|\sqrt{n_{k'}}}. \tag{10}$$

*Proof.* The following holds, and the result follows trivially from Proposition 2.

$$\max_{\Theta} \sum_{k\in\mathcal{Y}} \sum_{k'\neq k} \Pi(\theta_{k,k'};k) + \Pi(\theta_{k',k};k') \leq \sum_{k\in\mathcal{Y}} \sum_{k'\neq k} \max_{\theta_{k,k'},\theta_{k',k}} \Pi(\theta_{k,k'};k) + \Pi(\theta_{k',k};k'). \quad (49)$$

$\square$

### C.2 CASE OF RELU NEURAL NETWORKS

In this section, we specifically demonstrate the values of $\mathcal{C}(\mathcal{H}_l, \mathcal{X})$ and $B$ when the feature map is restricted to ReLU neural feature maps. This allows us to validate the assumption $\mathfrak{R}_{n_k}(\mathcal{H}_l) \leq \frac{\mathcal{C}(\mathcal{H}_l,\mathcal{X})}{\sqrt{n_k}}$ posed in this paper and to illustrate the practicality of the proposed theory.

Let $\overline{\mathcal{F}}$ represent the set of ReLU neural feature map, and define a ReLU neural feature map as $\overline{\boldsymbol{f}}(\boldsymbol{x}) = \boldsymbol{W}^1 \max(\boldsymbol{0}, \boldsymbol{W}^2 \ldots \max(\boldsymbol{0}, \boldsymbol{W}^q \boldsymbol{x}) \ldots) \in \overline{\mathcal{F}}$. Here, $q$ denotes the depth of the network, and the $\max$ operation is applied element-wise. We define the set $\overline{\mathcal{F}}^M$ as the set of ReLU neural feature map where the product of the Frobenius norms from the first layer to the second-to-last layer is bounded by $M$. That is,

$$\overline{\mathcal{F}}^M = \left\{ \overline{\boldsymbol{f}}' : \boldsymbol{x} \mapsto \boldsymbol{W}^1 \max(\boldsymbol{0}, \boldsymbol{W}^2 \ldots \max(\boldsymbol{0}, \boldsymbol{W}^q \boldsymbol{x}) \ldots) \in \overline{\mathcal{F}} \mid \prod_{s=2}^{q} \|\boldsymbol{W}^s\|_F \leq M \right\}. \quad (50)$$

We focus on functions that belong to $\overline{\mathcal{F}}^M$. When discussing ReLU neural networks, we add an overline to distinguish them from the general case. For instance, $\overline{\boldsymbol{h}}$ refers to a function that returns the output of the second-to-last layer of a ReLU neural feature map. Similarly, $\overline{\boldsymbol{\mu}}(\mathcal{S}_k)$ represents the average of the features $\overline{\boldsymbol{f}}(\boldsymbol{x})$ for the dataset $\mathcal{S}_k$.

For such networks, we similarly define $\overline{\mathcal{H}}_l^M$, their Rademacher complexity, and empirical Rademacher complexity as follows:

$$\overline{\mathcal{H}}_l^M \equiv \{(\boldsymbol{v}_l, \overline{\boldsymbol{h}}) \mid \overline{\boldsymbol{f}} \in \overline{\mathcal{F}}^M, \overline{\boldsymbol{f}} = \boldsymbol{W}^1 \overline{\boldsymbol{h}}, \ \boldsymbol{v}_l \text{ is the } l\text{-th right singular vector of } \boldsymbol{W}^1\}, \quad (51)$$

$$\mathfrak{R}_{n_k}\left(\overline{\mathcal{H}}_l^M\right) \equiv \mathbb{E}_{\mathcal{S}_k' \sim P_k^{n_k}, \boldsymbol{\sigma}} \left[ \sup_{(\boldsymbol{v}_l, \overline{\boldsymbol{h}}) \in \overline{\mathcal{H}}_l^M} \frac{1}{n_k} \sum_{\boldsymbol{x}_j \in \mathcal{S}_k'} \sigma_j \boldsymbol{v}_l^\top \overline{\boldsymbol{h}}(\boldsymbol{x}_j) \right], \quad (52)$$

$$\hat{\mathfrak{R}}_{\mathcal{S}_k}\left(\overline{\mathcal{H}}_l^M\right) \equiv \mathbb{E}_{\boldsymbol{\sigma}} \left[ \sup_{(\boldsymbol{v}_l, \overline{\boldsymbol{h}}) \in \overline{\mathcal{H}}_l^M} \frac{1}{n_k} \sum_{\boldsymbol{x}_j \in \mathcal{S}_k} \sigma_j \boldsymbol{v}_l^\top \overline{\boldsymbol{h}}(\boldsymbol{x}_j) \right]. \quad (53)$$

According to Golowich et al. (2018), the empirical Rademacher complexity for this family of functions is bounded by:

$$\begin{aligned} \hat{\mathfrak{R}}_{\mathcal{S}_k}\left(\overline{\mathcal{H}}_l^M\right) &= \mathbb{E}_{\boldsymbol{\sigma}} \left[ \sup_{(\boldsymbol{v}_l, \overline{\boldsymbol{h}}) \in \overline{\mathcal{H}}_l^M} \frac{1}{n_k} \sum_{\boldsymbol{x}_j \in \mathcal{S}_k} \sigma_j \boldsymbol{v}_l^\top \overline{\boldsymbol{h}}(\boldsymbol{x}_j) \right] \\ &\leq \frac{1}{\sqrt{n_k}} \left( \sqrt{2q \log 2} + 1 \right) M \sup_{\boldsymbol{x} \in \mathcal{X}} \|\boldsymbol{x}\| \\ &\leq \frac{1}{\sqrt{n_k}} (1.5\sqrt{q} + 1) M \sup_{\boldsymbol{x} \in \mathcal{X}} \|\boldsymbol{x}\|. \end{aligned} \quad (54)$$

Taking the expectation over $\mathcal{S}_k$, we can replace $\bar{\mathcal{C}}(\overline{\mathcal{F}}, \mathcal{X}) = (1.5\sqrt{q} + 1) M \sup_{\boldsymbol{x} \in \mathcal{X}} \|\boldsymbol{x}\|$. Since $\bar{\mathcal{C}}(\overline{\mathcal{F}}, \mathcal{X})$ does not depend on $n_k$, this confirms the assumption. We can also replace $B = M \sup_{\boldsymbol{x} \in \mathcal{X}} \|\boldsymbol{x}\|$ because $\sup_{\boldsymbol{x} \in \mathcal{X}, \boldsymbol{h} \in \mathcal{H}} \|\boldsymbol{h}(\boldsymbol{x})\| \leq M \sup_{\boldsymbol{x} \in \mathcal{X}} \|\boldsymbol{x}\|$.

In this case, Lemmas 3 and 4 can be rewritten as follows. Since these are simple substitutions, we omit the proofs.

**Lemma 5.** *Suppose $n_k > 2$. For any $\overline{\boldsymbol{f}} \in \overline{\mathcal{F}}^M$ and any $\delta > 0$, the following holds with a probability at least $1 - \delta$:*

$$\mathbb{E}_{\boldsymbol{x},\boldsymbol{x}'\sim P_k}\left[\left\|\overline{\boldsymbol{f}}(\boldsymbol{x}) - \overline{\boldsymbol{f}}(\boldsymbol{x}')\right\|\right] - \|\boldsymbol{W}^1\|_2 \sum_{l=1}^{r} \mathrm{Avg}_{\boldsymbol{x},\boldsymbol{x}'\in\mathcal{S}_k}\left|\boldsymbol{v}_l^\top\left(\overline{\boldsymbol{h}}(\boldsymbol{x}) - \overline{\boldsymbol{h}}(\boldsymbol{x}')\right)\right|$$

$$< \frac{r\|\boldsymbol{W}^1\|_2 M \sup_{\boldsymbol{x}\in\mathcal{X}} \|\boldsymbol{x}\|}{\sqrt{n_k}}\left(6\sqrt{q} + 8 + \sqrt{2\log\frac{r}{\delta}}\right). \quad (55)$$

**Lemma 6.** *Suppose $n_k > 2$. For $\overline{\boldsymbol{f}} \in \overline{\mathcal{F}}^M$, assume that for all $\boldsymbol{x} \in \mathcal{S}_k$, $\overline{\boldsymbol{f}}(\boldsymbol{x}) = \overline{\boldsymbol{\mu}}(\mathcal{S}_k)$ holds. For any $0 < \delta, \delta' < 1$, define $\theta$ as follows:*

$$\theta \equiv \frac{\pi}{2\delta'} \frac{r\|\boldsymbol{W}^1\|_2 M \sup_{\boldsymbol{x}\in\mathcal{X}} \|\boldsymbol{x}\|}{\sqrt{n_k}\|\overline{\boldsymbol{\mu}}(\mathcal{S}_k)\|}\left(6\sqrt{q} + 8 + \sqrt{2\log\frac{r}{\delta}}\right). \quad (56)$$

*Assume that $\delta, \delta'$ satisfy $\theta \leq \frac{\pi}{2}$. For $\boldsymbol{x} \sim P_k$, the probability that the angle between $\overline{\boldsymbol{f}}(\boldsymbol{x})$ and $\overline{\boldsymbol{\mu}}(\mathcal{S}_k)$ is less than $\theta$ is at least $1 - (\delta + \delta')$.*

We define the angular bound probability $\overline{\Pi}(\theta; k)$ for a ReLU neural network as in Section 4.2. Then, Proposition 1 can be rewritten as follows. The proof is omitted here as well.

**Lemma 7.** *Suppose $n_k > 2$. For $\overline{\boldsymbol{f}} \in \overline{\mathcal{F}}^M$, assume that for all $\boldsymbol{x} \in \mathcal{S}_k$, $\overline{\boldsymbol{f}}(\boldsymbol{x}) = \overline{\boldsymbol{\mu}}(\mathcal{S}_k)$ holds. Consider any $\theta$ satisfying the following conditions:*

$$\frac{\pi r\|\boldsymbol{W}^1\|_2 M \sup_{\boldsymbol{x}\in\mathcal{X}} \|\boldsymbol{x}\|}{2\sqrt{n_k}\|\overline{\boldsymbol{\mu}}(\mathcal{S}_k)\|}\left(6\sqrt{q} + 8 + \sqrt{2\log\left(\frac{\sqrt{n_k}\|\overline{\boldsymbol{\mu}}(\mathcal{S}_k)\|}{\|\boldsymbol{W}^1\|_2}\right)}\right) < \theta < \frac{\pi}{2}. \quad (57)$$

*For such $\theta$, the following holds:*

$$\overline{\Pi}(\theta; k) = 1 - \frac{\pi r\|\boldsymbol{W}^1\|_2}{2\theta\sqrt{n_k}\|\overline{\boldsymbol{\mu}}(\mathcal{S}_k)\|}\left(1 + M \sup_{\boldsymbol{x}\in\mathcal{X}} \|\boldsymbol{x}\|\left(6\sqrt{q} + 8 + \sqrt{2\log\left(\frac{2\theta\sqrt{n_k}\|\overline{\boldsymbol{\mu}}(\mathcal{S}_k)\|}{\pi\|\boldsymbol{W}^1\|_2}\right)}\right)\right). \quad (58)$$

Propositions 2 and 3 also hold similarly, ensuring the validity of 1vs1adjuster.

## C.3 MLA Approximates 1vs1adjuster

In this section, we provide the proof of Lemma 2.

**Lemma 2.** *Let $\theta \in [0, 1)$. Then, the following holds:*

$$\phi(\theta) \leq \tan\left(\frac{\pi}{2}\theta\right) < \frac{\pi}{2}\phi(\theta) \qquad \left(0 \leq \theta \leq \frac{1}{2}\right), \quad (18)$$

$$\frac{2}{\pi}\phi(\theta) < \tan\left(\frac{\pi}{2}\theta\right) < \phi(\theta) \qquad \left(\frac{1}{2} < \theta < 1\right). \quad (19)$$

*Proof.* When $\theta = 0$, the result is trivial since $\phi(0) = \tan(0) = 0$. For the remainder, we assume $\theta \in (0, 1)$ unless otherwise stated. Define $h(\theta) \equiv \frac{\tan\left(\frac{\pi}{2}\theta\right)}{\phi(\theta)} = \frac{\tan\left(\frac{\pi}{2}\theta\right)(1-\theta)}{\theta}$. Since $h(\theta) = \frac{1}{h(1-\theta)}$, proving the following will conclude the proof:

$$1 \leq h(\theta) < \frac{\pi}{2} \qquad \left(0 < \theta \leq \frac{1}{2}\right). \quad (59)$$

First, we demonstrate that $h(\theta)$ is monotonically non-increasing.

$$h'(\theta) = \frac{\theta\left(\frac{\pi}{2}\frac{1}{\cos^2\left(\frac{\pi}{2}\theta\right)}(1-\theta) - \tan\left(\frac{\pi}{2}\theta\right)\right) - \tan\left(\frac{\pi}{2}\theta\right)(1-\theta)}{\theta^2}$$

$$= \frac{\frac{\pi}{2}\frac{\theta(1-\theta)}{\cos^2\left(\frac{\pi}{2}\theta\right)} - \tan\left(\frac{\pi}{2}\theta\right)}{\theta^2}. \quad (60)$$

Now, define $l(\theta) \equiv \frac{\pi}{2} \frac{\theta(1-\theta)}{\cos^2\left(\frac{\pi}{2}\theta\right)} - \tan\left(\frac{\pi}{2}\theta\right)$ for $\theta \in [0,1)$:

$$
\begin{aligned}
l'(\theta) &= \frac{\pi}{2}\left(\frac{(-2\theta+1)\cos^2\left(\frac{\pi}{2}\theta\right) - \pi\theta(1-\theta)\sin(\frac{\pi}{2}\theta)\cos(\frac{\pi}{2}\theta)}{\cos^4\left(\frac{\pi}{2}\theta\right)} - \frac{1}{\cos^2\left(\frac{\pi}{2}\theta\right)}\right) \\
&= -\frac{\pi}{2}\frac{2\theta\cos^2\left(\frac{\pi}{2}\theta\right) + \pi\theta(1-\theta)\sin(\frac{\pi}{2}\theta)\cos(\frac{\pi}{2}\theta)}{\cos^4\left(\frac{\pi}{2}\theta\right)} \\
&\leq 0. \hspace{9cm} (61)
\end{aligned}
$$

Since $l(0) = 0$, we have $l(\theta) \leq 0$ for $\theta \in [0,1)$. Therefore, from Eq. (60), $h'(\theta) \leq 0$ for $\theta \in (0,1)$, indicating that $h(\theta)$ is monotonically non-increasing.

Since $h(\theta) = 1$, to prove Eq. (59), we must indicate that $\lim_{\theta \to +0} h(\theta) = \frac{\pi}{2}$. By applying l'Hôpital's rule, this can be verified.

$$
\begin{aligned}
\lim_{\theta \to +0} h(\theta) &= \lim_{\theta \to +0} \frac{\tan\left(\frac{\pi}{2}\theta\right)(1-\theta)}{\theta} \\
&= \lim_{\theta \to +0} -\tan\left(\frac{\pi}{2}\theta\right) + \frac{\pi}{2}\frac{(1-\theta)}{\cos^2\left(\frac{\pi}{2}\theta\right)} \\
&= \frac{\pi}{2}. \hspace{8cm} (62)
\end{aligned}
$$

Thus, Eq. (59) holds, completing the proof.

$\square$

## D  Algorithm of 1vs1adjuster

Algorithm 1 outlines the detailed steps of 1vs1adjuster, a 1-vs-1 multi-class classifier that performs classification based on the decision boundaries proposed in Proposition 3. The derivation of the normal vector $\boldsymbol{m}_{k,k'} \in \mathbb{R}^d$ for the decision boundary between $k$ and $k'$ is as follows. Since $\boldsymbol{m}_{k,k'}$ lies in the same plane as $\boldsymbol{w}_k$ and $\boldsymbol{w}_{k'}$, and the angles with each are specified, it can be expressed using $\alpha, \beta \in \mathbb{R}$ as follows. Note that we assume $\|\boldsymbol{m}_{k,k'}\| = 1$ here.

$$
\begin{cases}
\boldsymbol{m}_{k,k'} = \alpha\boldsymbol{w}_k + \beta\boldsymbol{w}_{k'} \\
\boldsymbol{m}_{k,k'}^\top \boldsymbol{w}_k = \cos\left(\frac{\pi}{2} - \theta_{k,k'}^*(\gamma_{1\mathrm{v}1})\right) \\
\boldsymbol{m}_{k,k'}^\top \boldsymbol{w}_{k'} = \cos\left(\frac{\pi}{2} + \theta_{k',k}^*(\gamma_{1\mathrm{v}1})\right)
\end{cases}
\tag{63}
$$

Solving this system of equations for $\alpha$ and $\beta$, we obtain the following:

$$
\begin{cases}
\alpha = \dfrac{\sin\theta_{k,k'}^*(\gamma_{1\mathrm{v}1}) + \sin\theta_{k',k}^*(\gamma_{1\mathrm{v}1})\cos\psi}{1 - \cos^2\psi} \\
\beta = -\dfrac{\sin\theta_{k',k}^*(\gamma_{1\mathrm{v}1}) + \sin\theta_{k,k'}^*(\gamma_{1\mathrm{v}1})\cos\psi}{1 - \cos^2\psi}
\end{cases}
\tag{64}
$$

Thus, the decision boundary can be defined as a plane passing through the origin with $\boldsymbol{m}_{k,k'}$ as the normal vector. Note that Algorithm 1 simplifies this by using the normal vector multiplied by $1 - \cos^2\psi$. Additionally, when $K$ is sufficiently large, $\boldsymbol{m}_{k,k'}$ can be approximated as $\boldsymbol{m}_{k,k'} \sim \sin\theta_{k,k'}^*(\gamma_{1\mathrm{v}1})\boldsymbol{w}_k - \sin\theta_{k',k}^*(\gamma_{1\mathrm{v}1})\boldsymbol{w}_{k'}$.

## E  Discussion: Approximation of ALA

Similar to the approximation of MLA in Section 4.3, we explore approximating ALA to 1vs1adjuster. ALA modifies the logits by adding a correction term, $-\gamma_+ \log n_k$, where $\gamma_+ > 0$.

First, we assume that for any $\boldsymbol{x} \in \mathcal{X}$, the feature norm is constant, i.e., $\boldsymbol{f}(\boldsymbol{x}) = |\boldsymbol{f}|$. Similar to the MLA case, we denote the coefficient of the ALA by $\eta_k$. To adjust the decision boundaries in

---

**Algorithm 1** 1vs1adjuster

---

1: **function** ONE_VS_ONE($\boldsymbol{f}(\boldsymbol{x}), \boldsymbol{W}, \Theta^*_{\gamma_{1v1}}$)
2:     **let** $counter[1..K]$ be a zero-initialized array
3:     **for all** $k \leftarrow \mathcal{Y}$ **do**
4:         **for all** $k' \leftarrow \mathcal{Y}$ **do**
5:             **if** $k = k'$ **then**
6:                 **continue**
7:             **end if**
8:             **let** $\alpha = \sin\theta^*_{k,k'}(\gamma_{1v1}) + \sin\theta^*_{k',k}(\gamma_{1v1})\cos\psi$
9:             **let** $\beta = \sin\theta^*_{k',k}(\gamma_{1v1}) + \sin\theta^*_{k,k'}(\gamma_{1v1})\cos\psi$
10:             **let** $\boldsymbol{m}_{k,k'} = \alpha\boldsymbol{w}_k + \beta\boldsymbol{w}_{k'}$
11:             **if** $\boldsymbol{m}_{k,k'}^\top \boldsymbol{f}(\boldsymbol{x}) > 0$ **then**
12:                 $counter[k] \leftarrow counter[k] + 1$
13:             **end if**
14:         **end for**
15:     **end for**
16:     **return** $\arg\max(counter)$
17: **end function**

---

ALA such that it satisfies Proposition 3, $\eta_k$ must be set so that the following condition holds for all $k \neq k' \in \mathcal{Y}$:

$$\|\boldsymbol{f}\|\cos(\theta^*_{k,k'}(\gamma_{1v1})) - \eta_k = \|\boldsymbol{f}\|\cos(\theta^*_{k',k}(\gamma_{1v1})) - \eta_{k'}$$
$$\Leftrightarrow \|\boldsymbol{f}\|\big(\cos(\theta^*_{k,k'}(\gamma_{1v1})) - \cos(\theta^*_{k',k}(\gamma_{1v1}))\big) = \eta_k - \eta_{k'}. \tag{65}$$

For the left-hand side:

$$\cos(\theta^*_{k,k'}(\gamma_{1v1})) - \cos(\theta^*_{k',k}(\gamma_{1v1}))$$
$$= -2\sin\left(\frac{\theta^*_{k,k'}(\gamma_{1v1}) + \theta^*_{k',k}(\gamma_{1v1})}{2}\right)\sin\left(\frac{\theta^*_{k,k'}(\gamma_{1v1}) - \theta^*_{k',k}(\gamma_{1v1})}{2}\right)$$
$$= -2\sin\left(\frac{\psi}{2}\right)\sin\left(\frac{\psi}{2} - \theta^*_{k',k}(\gamma_{1v1})\right)$$
$$= -2\sin\left(\frac{\psi}{2}\right)\sin\left(\theta^*_{k,k'}(\gamma_{1v1}) - \frac{\psi}{2}\right)$$
$$= 2\sin\left(\frac{\psi}{2}\right)\sin\left(\frac{\psi}{2} - \theta^*_{k,k'}(\gamma_{1v1})\right) \tag{66}$$
$$= 2\sin\left(\frac{\psi}{2}\right)\sin\left(\frac{\psi}{2}\left(1 - \frac{2}{\tau_{k,k'}(\gamma_{1v1}) + 1}\right)\right).$$

Let us denote the right-hand side as $g(\tau_{k,k'}(\gamma_{1v1}))$. We set $\gamma^*_+ > 0$ so that it can be approximated near $\tau_{k,k'}(\gamma_{1v1}) = 1$ by $2\gamma^*_+ \log(\tau_{k,k'}(\gamma_{1v1}))$. Specifically, we set $\gamma^*_+$ to satisfy:

$$g(1) = 0 = -2\gamma^*_+ \log(1), \tag{67}$$
$$g'(1) = \frac{\psi}{2}\sin\left(\frac{\psi}{2}\right) = 2\gamma^*_+. \tag{68}$$

The solution is $\gamma^*_+ = \frac{\psi}{4}\sin(\frac{\psi}{2})$. Thus, when $\tau_{k,k'}(\gamma_{1v1}) \sim 1$:

$$\eta_k - \eta_{k'} \sim 2\gamma^*_+ \log\tau_{k,k'}(\gamma_{1v1})$$
$$= \gamma^*_+\big(\log(\|\boldsymbol{\mu}(\mathcal{S}_k)\|^2 n_k^{\gamma_{1v1}}) - \log(\|\boldsymbol{\mu}(\mathcal{S}_{k'})\|^2 n_{k'}^{\gamma_{1v1}})\big)$$
$$= \gamma^*_+(\gamma_{1v1}\log n_k - \gamma_{1v1}\log n_{k'} + 2(\log\|\boldsymbol{\mu}(\mathcal{S}_k)\| - \log\|\boldsymbol{\mu}(\mathcal{S}_{k'})\|))$$
$$= \gamma^*_+\gamma_{1v1}(\log n_k - \log n_{k'}). \tag{69}$$

From this, we conclude that when $\tau_{k,k'}(\gamma_{1v1}) \sim 1$ and $\|\boldsymbol{\mu}(\mathcal{S}_k)\| = \|\boldsymbol{\mu}(\mathcal{S}_{k'})\|$ for all $k, k' \in \mathcal{Y}$, the ALA results in the decision boundaries as 1vs1adjuster. However, under LTR settings, $\tau_{k,k'}(\gamma_{1v1})$ can vary significantly depending on $k$ and $k'$, making this approximation less realistic in practice.

# F  EXPERIMENTS

In this section, we summarize the details of the experimental settings and results that are not fully covered in the main text. Appendix F.1 outlines the detailed experimental settings. Appendix F.2 provides supplementary results for the experiments in Section 5.2, while Appendix F.3 presents additional results for the experiments in Section 5.3 under different conditions.

## F.1  SETTINGS

We describe the detailed training settings in this section. We mainly followed the hyperparameters used in Hasegawa & Sato (2023). First, we provide the details for experiments on image datasets, and then describe the experiments on tabular data, highlighting the differences.

### F.1.1  DATASETS

Following Cui et al. (2019) and Liu et al. (2019), we created long-tailed versions of image datasets. For tuning the hyperparameters $\gamma_{1v1}, \gamma_+, \gamma_\times$, we used validation datasets. Since CIFAR10 and CIFAR100 do not have validation datasets, we created validation datasets using a portion of the training datasets. Following Liu et al. (2019), we constructed the validation datasets by extracting only 20 samples per class from the training datasets and using the remaining samples as the training datasets. For CIFAR100, we set $n_1$ to $480$, and for CIFAR10, we set it to $4980$. The imbalance factor $\rho$ was set to $100$. The classes within each dataset were divided into three groups—*Many*, *Medium*, and *Few*—on the basis of the number of training samples $n_k$. For CIFAR100-LT and ImageNet-LT, we categorized classes $k$ as *Many* if they had more than 1000 training samples ($n_k > 1000$), *Medium* if they had between 200 and 1000 training samples ($200 \le n_k \le 1000$), and *Few* otherwise. For CIFAR10-LT, we classified classes as *Many* if they had more than 100 samples, *Medium* if they had between 20 and 100 samples, and *Few* otherwise.

### F.1.2  EVALUATION METRICS

Unless otherwise specified, we used the following hyperparameters with ResNet. We chose stochastic gradient descent with momentum $= 0.9$ as the optimizer and applied a cosine learning rate scheduler (Loshchilov & Hutter, 2017) to gradually decrease the learning rate from $0.01$ to $0$. The batch size was set to $64$, and the number of training epochs was $320$. The loss function used was cross-entropy loss, and regularization included a weight decay of $0.005$ (Hanson & Pratt, 1989) and feature regularization of $0.01$ (Hasegawa & Sato, 2023). Although we used an ETF classifier for the linear layer, we did not use Dot-Regression Loss (Yang et al., 2022b), following Hasegawa & Sato (2023). The optimal $\gamma \in \{\gamma_{1v1}, \gamma_+, \gamma_\times\}$ for LA and 1vs1adjuster were determined using cross-validation on validation datasets, exploring values from $\{0.00, 0.05, \dots, 2.00\}$.

Next, we describe the experimental setup for ImageNet-LT. The learning rate was gradually decreased from $0.05$ to $0$, with the number of training epochs set to $200$. Regularization involved a weight decay of $0.00024$ and a feature regularization of $0.00003$. All other settings were the same as those used for CIFAR100-LT.

For the experiments in Section 5.2, we used the optimal values of $\gamma_{1v1}, \gamma_+, \gamma_\times$ determined by the validation data. For ALA, the value of $\|\boldsymbol{f}\|$ was calculated by averaging the norm of the class mean features for each class.

The accuracy reported in Section 5.3 represents the mean and standard deviation across five independent experiments, each using different random seeds. The values of $\gamma_{1v1}$, $\gamma_\times$, and the training accuracy mentioned in Section 5.4 correspond to these experiments. All experiments were conducted on a single NVIDIA A100.

### F.1.3  TABULAR DATA

Next, we describe the experiments conducted on tabular data. Since tabular data have characteristics that differ significantly from image data, we conducted these experiments to demonstrate that our results can generalize to other modalities beyond images. Apart from the settings described below, the procedure was identical to that used for image data. Following Hasegawa & Sato (2023), we used

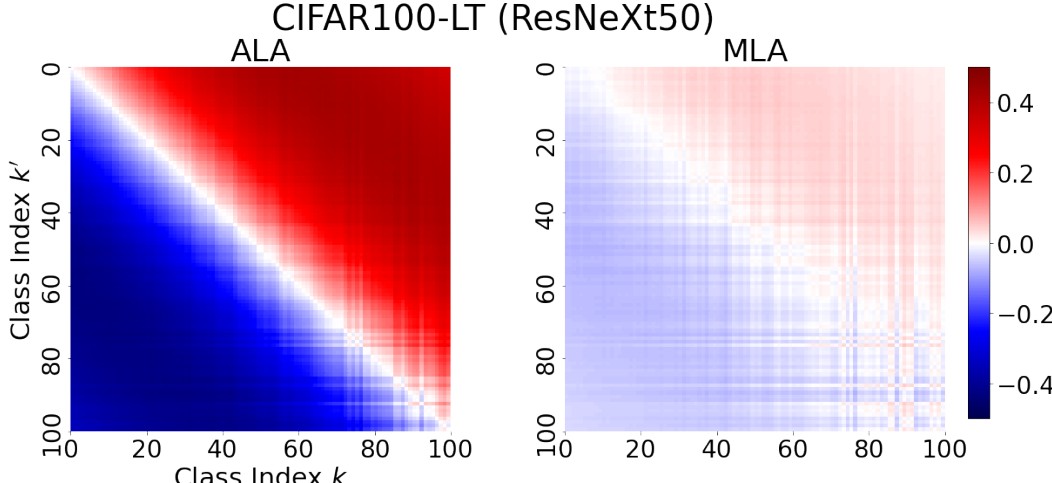

Figure 4: Heatmaps showing the difference in angles between the decision boundaries adjusted by each method and those adjusted by 1vs1adjuster of ResNeXt50 trained on CIFAR100-LT. The angle differences between MLA and 1vs1adjuster are generally small compared to the difference between ALA and 1vs1adjuster.

the Helena dataset with 100 classes. Since this dataset is not pre-split into validation and test sets, we randomly sampled 20 non-overlapping samples per class for validation and test sets respectively. The distribution of this dataset resembles a long-tailed distribution with $\rho \simeq 40$. For each class, those with more than 500 training samples were categorized as *Many*, those with $200 \leq n_k \leq 500$ as *Medium*, and the rest as *Few*.

In line with Kadra et al. (2021), we trained a MLP with sufficient regularization. We used a 9-layer MLP with 512-dimensional hidden layers as the feature map. The model was trained for 400 epochs using AdamW (Loshchilov & Hutter, 2018). Regularization methods included a $0.15$ dropout rate (Srivastava et al., 2014), $0.15$ weight decay, and $0.001$ feature regularization.

### F.2    DECISION BOUNDARY ANGLES

We present the results of the experiments from Section 5.2 conducted with different models and datasets. Figure 4 illustrates the differences in the angles of the decision boundaries of ResNeXt50 trained on CIFAR100-LT, while Figures 5 and 6 display the results for ImageNet-LT and Helena, respectively. In all cases, the decision boundaries adjusted by MLA and 1vs1adjuster tend to be more similar to each other than to those adjusted by ALA.

### F.3    TEST ACCURACY

In this subsection, we first present the error bar plot for the average test accuracy on Helena, which is not shown in Section 5.3, followed by a detailed table of test accuracy of models adjusted by each method.

**Error Bar**    Figure 7 shows the average accuracy of models trained on Helena, adjusted by different methods. Notably, models trained on Helena, similar to those on ImageNet-LT, did not achieve a training accuracy of $100\%$, indicating that NC has not fully occurred (see Table 1). Despite this, the results show that MLA and 1vs1adjuster achieve comparable accuracy, consistent with the experiments on other image datasets.

**Table**    We compare the test accuracy when each method is applied post-hoc to trained models. We refer to the models only trained on the training data, without the application of any post-hoc

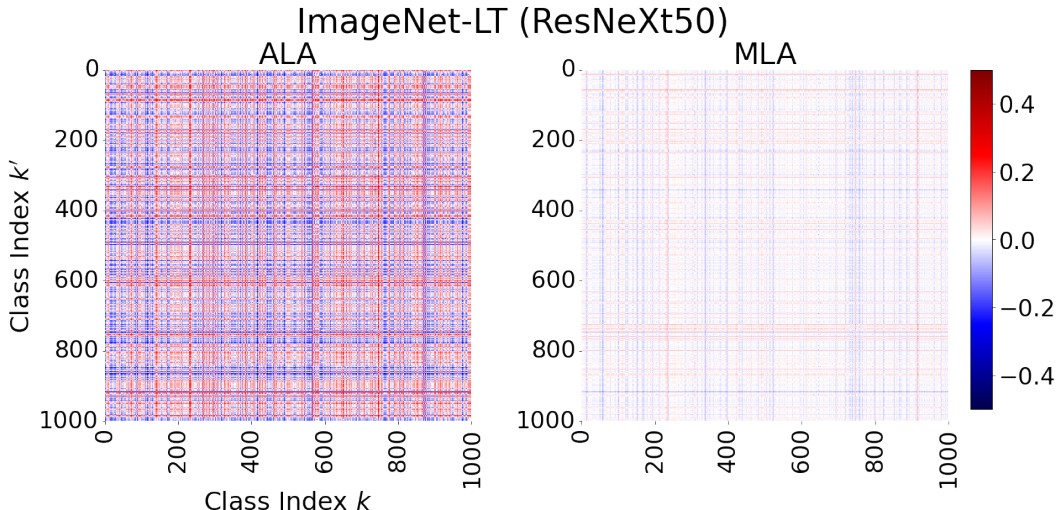

Figure 5: Heatmaps showing the difference in angles between the decision boundaries adjusted by each method and those adjusted by 1vs1adjuster of ResNeXt50 trained on ImageNet-LT. The angle differences between MLA and 1vs1adjuster are generally small compared to the difference between ALA and 1vs1adjuster.

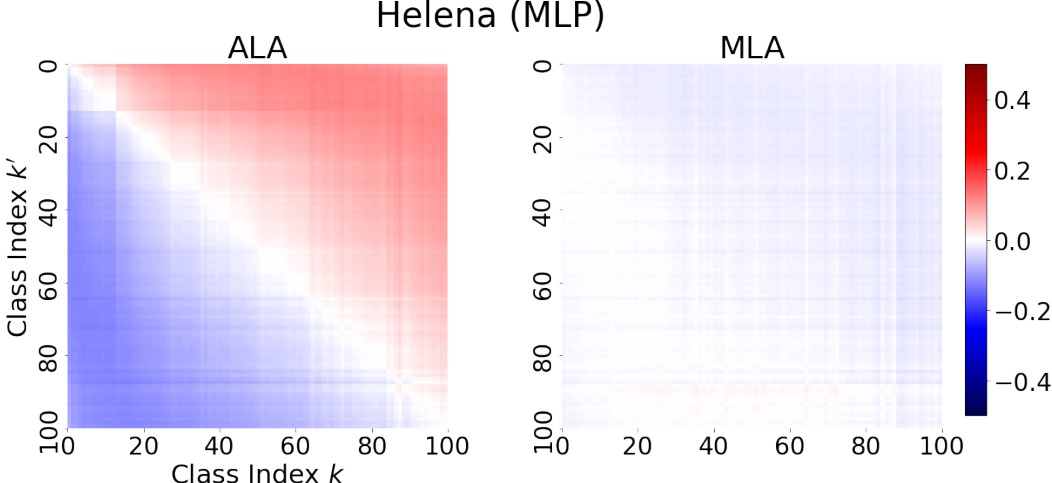

Figure 6: Heatmaps showing the difference in angles between the decision boundaries adjusted by each method and those adjusted by 1vs1adjuster of MLP trained on Helena. The angle differences between MLA and 1vs1adjuster are generally small compared to the difference between ALA and 1vs1adjuster.

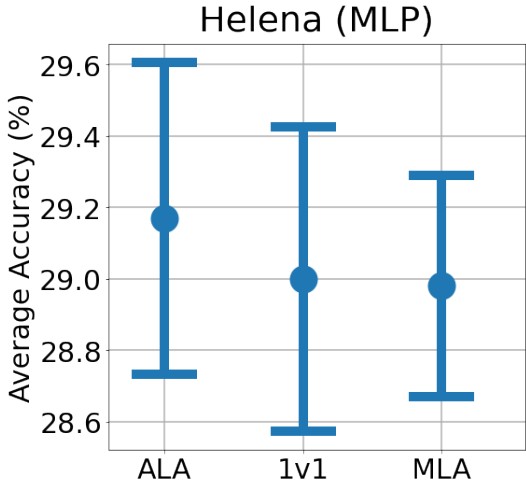

Figure 7: Average accuracy of MLP trained on Helena and adjusted by different methods. MLA and 1vs1adjuster achieve comparable accuracy as with the experiments on other datasets in Figure 3.

Table 6: Accuracy of ResNet34 adjusted with each method on CIFAR100-LT. MLA and 1vs1adjuster achieve comparable average accuracy. MLA outperforms ALA in average accuracy on CIFAR100-LT.

| Method | *Many* | *Medium* | *Few* | Average |
|---|---|---|---|---|
| Baseline | **77.9**$_{\pm\mathbf{0.3}}$ | $46.8_{\pm1.0}$ | $15.3_{\pm0.3}$ | $47.6_{\pm0.5}$ |
| ALA ($\gamma_+ = 1.0$) | $72.3_{\pm0.2}$ | $47.9_{\pm1.1}$ | $28.3_{\pm0.9}$ | $50.1_{\pm0.4}$ |
| ALA (best) | $75.6_{\pm0.9}$ | $49.2_{\pm1.0}$ | $25.2_{\pm2.2}$ | $50.7_{\pm0.4}$ |
| 1v1 ($\gamma_{1v1} = 0.5$) | $76.4_{\pm0.3}$ | **51.7**$_{\pm\mathbf{0.9}}$ | $24.7_{\pm0.7}$ | $51.7_{\pm0.2}$ |
| 1v1 (best) | $74.2_{\pm1.0}$ | **53.0**$_{\pm\mathbf{0.9}}$ | **29.6**$_{\pm\mathbf{1.1}}$ | **52.9**$_{\pm\mathbf{0.3}}$ |
| MLA ($\gamma_\times = 0.5$) | $75.6_{\pm0.3}$ | $52.3_{\pm0.8}$ | $27.1_{\pm0.7}$ | $52.4_{\pm0.2}$ |
| MLA (best) | $73.2_{\pm0.8}$ | **52.8**$_{\pm\mathbf{0.9}}$ | **30.9**$_{\pm\mathbf{1.1}}$ | **53.0**$_{\pm\mathbf{0.3}}$ |

adjustment methods, as Baseline. We present two cases for each adjustment method: when the hyperparameter values are set according to the values derived from each theory (0.5 or 1.0), and when the parameters are tuned using validation data (best). In addition to the overall average accuracy (Average), we also report the average accuracy for the *Many*, *Medium*, and *Few* categories. Refer to Appendix F.1 for details on these categorizations. The results for CIFAR100-LT and CIFAR10-LT are presented in Tables 6 and 7, respectively. Table 8 reports the accuracy of ResNeXt50 on CIFAR100-LT, while Tables 9 and 10 display the results of ImageNet-LT and Helena, respectively.

In all results, the Baseline shows higher accuracy for the *Many* category, as naive training on long-tailed data results in outputs biased toward head classes. The 1vs1adjuster and post-hoc LA methods successfully adjust for this bias by increasing the accuracy of *Few* classes, thereby improving the overall average accuracy. As shown in Section 5.3, MLA achieves nearly the same average accuracy as the 1vs1adjuster across all datasets. Additionally, MLA outperforms ALA in terms of average accuracy for CIFAR100-LT.

### F.4 COMBINATION WITH OTHER METHODS

Since MLA only adjusts the logits during inference, it can be seamlessly integrated with other LTR techniques. Notably, methods classified under "Information Augmentation" and "Module Improvement" in the taxonomy by Zhang et al. (2021) are particularly compatible. Here, we present experimental results combining MLA with the approach proposed by Wang et al. (2023a), which falls under "Information Augmentation." We compare the average accuracies of a model trained with cross entropy loss against one with MLA. The dataset used is CIFAR100-LT of our implementation,

Table 7: Accuracy of ResNet34 adjusted with each method on CIFAR10-LT. Although the number of classes is insufficient and the assumptions of our theory do not hold completely, MLA and 1vs1adjuster achieve comparable average accuracy.

| Method | *Many* | *Medium* | *Few* | Average |
|---|---|---|---|---|
| Baseline | $89.2_{\pm 1.5}$ | $76.3_{\pm 2.5}$ | $60.8_{\pm 4.6}$ | $76.8_{\pm 0.8}$ |
| ALA ($\gamma_+ = 1.0$) | $87.2_{\pm 2.5}$ | $78.5_{\pm 2.9}$ | $71.7_{\pm 3.9}$ | $79.9_{\pm 1.0}$ |
| ALA (best) | $87.7_{\pm 1.2}$ | $78.2_{\pm 3.0}$ | $70.5_{\pm 2.7}$ | $79.7_{\pm 1.1}$ |
| 1v1 ($\gamma_{1v1} = 0.5$) | $88.0_{\pm 2.2}$ | $78.6_{\pm 2.7}$ | $68.9_{\pm 4.2}$ | $79.5_{\pm 0.8}$ |
| 1v1 (best) | $87.9_{\pm 1.8}$ | $79.3_{\pm 4.0}$ | $69.0_{\pm 2.7}$ | $79.6_{\pm 1.1}$ |
| MLA ($\gamma_\times = 0.5$) | $88.1_{\pm 2.2}$ | $78.7_{\pm 2.7}$ | $68.4_{\pm 4.3}$ | $79.4_{\pm 0.8}$ |
| MLA (best) | $88.0_{\pm 1.6}$ | $79.2_{\pm 3.7}$ | $68.8_{\pm 2.9}$ | $79.6_{\pm 1.1}$ |

Table 8: Accuracy of ResNeXt50 adjusted with each method on Cifar100-LT. MLA and 1vs1adjuster achieve comparable average accuracy. MLA outperforms ALA in average accuracy on CIFAR100-LT.

| Method | *Many* | *Medium* | *Few* | Average |
|---|---|---|---|---|
| Baseline | $77.4_{\pm 0.3}$ | $49.8_{\pm 0.5}$ | $18.2_{\pm 0.4}$ | $49.4_{\pm 0.3}$ |
| ALA ($\gamma_+ = 1.0$) | $73.3_{\pm 0.6}$ | $50.9_{\pm 0.5}$ | $31.0_{\pm 0.4}$ | $52.4_{\pm 0.3}$ |
| ALA (best) | $76.3_{\pm 0.3}$ | $51.9_{\pm 0.4}$ | $26.2_{\pm 0.4}$ | $52.2_{\pm 0.2}$ |
| 1v1 ($\gamma_{1v1} = 0.5$) | $75.6_{\pm 0.3}$ | $53.5_{\pm 0.5}$ | $27.6_{\pm 0.7}$ | $53.0_{\pm 0.2}$ |
| 1v1 (best) | $70.5_{\pm 0.7}$ | $54.0_{\pm 1.2}$ | $36.6_{\pm 0.9}$ | $54.2_{\pm 0.3}$ |
| MLA ($\gamma_\times = 0.5$) | $74.5_{\pm 0.3}$ | $53.7_{\pm 0.8}$ | $30.3_{\pm 0.7}$ | $53.5_{\pm 0.2}$ |
| MLA (best) | $72.9_{\pm 0.5}$ | $53.8_{\pm 1.1}$ | $33.4_{\pm 0.6}$ | $54.0_{\pm 0.3}$ |

Table 9: Accuracy of ResNeXt50 adjusted with each method on ImageNet-LT. Although insufficient NC has occurred (see Table 1) and the assumptions of our theory do not hold completely, MLA and 1vs1adjuster achieve comparable average accuracy.

| Method | *Many* | *Medium* | *Few* | Average |
|---|---|---|---|---|
| Baseline | $68.1_{\pm 0.2}$ | $43.0_{\pm 0.2}$ | $17.0_{\pm 0.5}$ | $49.1_{\pm 0.2}$ |
| ALA ($\gamma_+ = 1.0$) | $65.3_{\pm 0.2}$ | $48.1_{\pm 0.1}$ | $31.2_{\pm 0.7}$ | $52.4_{\pm 0.1}$ |
| ALA (best) | $63.1_{\pm 0.4}$ | $48.7_{\pm 0.2}$ | $36.7_{\pm 0.9}$ | $52.6_{\pm 0.2}$ |
| 1v1 ($\gamma_{1v1} = 0.5$) | $48.7_{\pm 0.5}$ | $48.0_{\pm 0.4}$ | $48.4_{\pm 0.6}$ | $48.3_{\pm 0.2}$ |
| 1v1 (best) | $62.8_{\pm 0.4}$ | $49.3_{\pm 0.3}$ | $34.7_{\pm 0.7}$ | $52.5_{\pm 0.2}$ |
| MLA ($\gamma_\times = 0.5$) | $38.9_{\pm 0.7}$ | $44.0_{\pm 0.3}$ | $52.3_{\pm 0.6}$ | $43.2_{\pm 0.3}$ |
| MLA (best) | $62.4_{\pm 0.4}$ | $49.3_{\pm 0.2}$ | $36.0_{\pm 0.5}$ | $52.6_{\pm 0.2}$ |

Table 10: Accuracy of MLP adjusted with each method on Helena. Although insufficient NC has occurred (see Table 1) and the assumptions of our theory do not hold completely, MLA and 1vs1adjuster achieve comparable average accuracy.

| Method | *Many* | *Medium* | *Few* | Average |
|---|---|---|---|---|
| Baseline | $36.1_{\pm 1.3}$ | $21.6_{\pm 0.6}$ | $17.4_{\pm 1.0}$ | $25.2_{\pm 0.3}$ |
| ALA ($\gamma_+ = 1.0$) | $34.7_{\pm 1.4}$ | $26.5_{\pm 0.8}$ | $23.8_{\pm 0.3}$ | $28.4_{\pm 0.3}$ |
| ALA (best) | $31.2_{\pm 1.2}$ | $28.8_{\pm 0.6}$ | $27.4_{\pm 0.3}$ | $29.2_{\pm 0.4}$ |
| 1v1 ($\gamma_{1v1} = 0.5$) | $32.9_{\pm 1.0}$ | $29.0_{\pm 1.0}$ | $24.7_{\pm 0.8}$ | $28.9_{\pm 0.4}$ |
| 1v1 (best) | $31.5_{\pm 1.2}$ | $29.8_{\pm 1.6}$ | $25.6_{\pm 0.5}$ | $29.0_{\pm 0.4}$ |
| MLA ($\gamma_\times = 0.5$) | $31.1_{\pm 0.9}$ | $30.2_{\pm 1.0}$ | $26.0_{\pm 0.6}$ | $29.1_{\pm 0.5}$ |
| MLA (best) | $31.3_{\pm 1.7}$ | $29.8_{\pm 1.8}$ | $25.8_{\pm 1.0}$ | $29.0_{\pm 0.3}$ |

Table 11: Accuracy of ResNet32 trained with Wang et al. (2023a) and adjusted with MLA on CIFAR100-LT. MLA can be used effectively in combination with other LTR methods to improve accuracy.

| Method | *Many* | *Medium* | *Few* | Average |
|---|---|---|---|---|
| Wang et al. (2023a) | **73.5** | 45.1 | 13.5 | 44.9 |
| Wang et al. (2023a) + MLA | 63.8 | **51.7** | **30.2** | **49.1** |

Table 12: Accuracy of ResNet34 adjusted with each method on CIFAR100-LT with $\rho = 200$. MLA and 1vs1adjuster achieve comparable average accuracy. MLA outperforms ALA in average accuracy on CIFAR100-LT.

| Method | *Many* | *Medium* | *Few* | Average |
|---|---|---|---|---|
| Baseline | **78.4**$_{\pm 0.3}$ | 48.4$_{\pm 0.5}$ | 12.5$_{\pm 0.5}$ | 43.1$_{\pm 0.3}$ |
| ALA ($\gamma_+ = 1.0$) | 69.9$_{\pm 0.8}$ | 46.3$_{\pm 0.7}$ | 21.4$_{\pm 1.0}$ | 43.4$_{\pm 0.7}$ |
| ALA (best) | 76.3$_{\pm 0.3}$ | 50.3$_{\pm 0.4}$ | 19.3$_{\pm 1.0}$ | 45.7$_{\pm 0.4}$ |
| 1v1 ($\gamma_{1v1} = 0.5$) | 76.5$_{\pm 0.4}$ | **53.3**$_{\pm 0.4}$ | 20.2$_{\pm 0.6}$ | 47.0$_{\pm 0.2}$ |
| 1v1 (best) | 74.6$_{\pm 0.8}$ | **53.6**$_{\pm 0.5}$ | **23.4**$_{\pm 1.2}$ | **47.8**$_{\pm 0.4}$ |
| MLA ($\gamma_\times = 0.5$) | 75.4$_{\pm 0.5}$ | 53.1$_{\pm 0.6}$ | 22.5$_{\pm 0.6}$ | 47.6$_{\pm 0.3}$ |
| MLA (best) | 73.5$_{\pm 0.9}$ | 52.9$_{\pm 1.0}$ | **24.4**$_{\pm 0.4}$ | 47.7$_{\pm 0.5}$ |

while other code and experimental settings adhere to the official implementation by Wang et al. (2023a). Note that these results are not directly comparable with our other experiments due to differences in setup. The results are summarized in Table 11. As shown, combining MLA with other LTR methods can further improve accuracy.

## F.5    HIGHER IMBALANCE FACTOR

To evaluate the effectiveness of MLA on more imbalanced datasets, we conducted experiments on the CIFAR-LT dataset with $\rho = 200$. All other experimental settings were identical to those described in Section 5.1. The results are shown in Tables 12 and 13. As in the case with $\rho = 100$, the results demonstrate that MLA closely approximates 1vs1adjuster in all scenarios. MLA achieves higher mean accuracy than ALA on CIFAR100-LT in this case as well.

Table 13: Accuracy of ResNet34 adjusted with each method on CIFAR10-LT with $\rho = 200$. MLA and 1vs1adjuster achieve comparable average accuracy.

| Method | *Many* | *Medium* | *Few* | Average |
|---|---|---|---|---|
| Baseline | **91.6**$_{\pm 1.6}$ | 69.7$_{\pm 2.2}$ | 54.0$_{\pm 3.3}$ | 70.0$_{\pm 0.6}$ |
| ALA ($\gamma_+ = 1.0$) | 81.7$_{\pm 8.1}$ | 69.3$_{\pm 2.4}$ | 69.3$_{\pm 4.6}$ | 73.0$_{\pm 1.8}$ |
| ALA (best) | 89.2$_{\pm 1.9}$ | 70.5$_{\pm 2.0}$ | 64.5$_{\pm 3.0}$ | 73.7$_{\pm 0.6}$ |
| 1v1 ($\gamma_{1v1} = 0.5$) | 88.4$_{\pm 3.2}$ | 71.4$_{\pm 2.0}$ | 64.9$_{\pm 4.6}$ | 73.9$_{\pm 0.6}$ |
| 1v1 (best) | 87.5$_{\pm 3.9}$ | **71.7**$_{\pm 1.9}$ | **66.1**$_{\pm 5.4}$ | **74.2**$_{\pm 0.7}$ |
| MLA ($\gamma_\times = 0.5$) | 88.8$_{\pm 3.0}$ | 71.4$_{\pm 2.0}$ | 64.1$_{\pm 4.5}$ | 73.7$_{\pm 0.7}$ |
| MLA (best) | 88.0$_{\pm 3.4}$ | **71.7**$_{\pm 1.9}$ | 65.4$_{\pm 5.2}$ | 74.1$_{\pm 0.8}$ |

