# OpenReview forum: "Multiplicative Logit Adjustment Approximates Neural-Collapse-Aware Decision Boundary Adjustment"
_ICLR.cc/2025/Conference — ICLR 2025 Poster_

### Official Review · Reviewer_K7Rv · 2024-10-25

**Soundness:** 3
**Presentation:** 3
**Contribution:** 3
**Rating:** 6
**Confidence:** 2

**Summary:**

The paper addresses the problem of class imbalance in classification models and analyzes the effectiveness of the Multiplicative Logit Adjustment (MLA) method. The authors propose a theory on optimal decision boundaries based on neural collapse, demonstrating that MLA approximates these optimal boundaries. The paper is supported by experiments across various modalities, providing insights into MLA’s performance under different conditions.

**Strengths:**

* The problem of class imbalance is clearly motivated, and the investigation into MLA is a valuable contribution to this area of research.
* The mathematical framework is well-developed, offering a solid theoretical foundation for understanding the method.
* The analysis extends beyond image datasets to include other modalities, such as text, enhancing the paper's applicability.
* Extensive experiments are conducted, which reinforce the theoretical claims and provide a comprehensive evaluation of MLA’s performance.

**Weaknesses:**

* While the theoretical concepts are accurately communicated, the writing could be improved by breaking up overly long sentences to enhance readability (e.g., lines 098-104).
* The results in Tables 6-10 are intriguing, but more explanation is needed to clarify the implications of "Many," "Medium," and "Few" categories. Additionally, the reasons for the Baseline outperforming others in the "Many" category should be discussed in more depth.
* The descriptions for tables and figures are too brief, making it difficult for readers to grasp the full context of the results. Adding more detailed captions would help in understanding the findings.
* The paper contains numerous formal definitions and formulas, which can disrupt the reading flow, especially for readers who are not deeply familiar with this specific research area.

**Questions:**

1. Why did you primarily focus on ResNet models for the image analysis? Were there specific reasons for this choice over other architectures?
2. Why did you create imbalanced versions of balanced datasets instead of using inherently imbalanced datasets? For example, datasets that exhibit natural class imbalances could offer more realistic evaluations.
3. Although the paper focuses on post-hoc LA methods, did you compare the results with loss-function-based approaches? How does MLA's performance compare to other logit adjustment methods?

---

> ### Author Response · Authors · 2024-11-21
>
> We sincerely appreciate your thorough review of our paper and your positive feedback. Your insights have been invaluable in shaping the direction of our work. We are grateful for the time and effort you dedicated to providing constructive comments. We would like to address some of the points raised in your review and provide additional context or clarification that we believe could contribute to an even more favorable evaluation.
>
>
> > W1: While the theoretical concepts are accurately communicated, the writing could be improved by breaking up overly long sentences to enhance readability (e.g., lines 098-104).
>
> A1: We have split several long sentences in the paper to improve readability. For example, we have revised the sentence in question to describe the four NC phenomena using a bulleted list format as follows.
>
>
> > W2: The results in Tables 6-10 are intriguing, but more explanation is needed to clarify the implications of "Many," "Medium," and "Few" categories. Additionally, the reasons for the Baseline outperforming others in the "Many" category should be discussed in more depth.
>
> A2: The terms "Many," "Medium," and "Few" represent groupings of classes based on the sample numbers, and naively trained models (Baseline) tend to produce outputs biased toward "Many" classes (head classes). Detailed explanations of this grouping are already provided in Appendix F.1. Specifically, classes with a large number of samples are categorized as "Many," those with fewer samples as "Few," and the remainder as "Medium." In Tables 6–10, the "Many," "Medium," and "Few" categories indicate the mean accuracy restricted to each respective class group.
>
> The primary goal of LTR is to increase the overall average accuracy across all classes. Naive training methods result in biased outputs favoring head classes ("Many" classes), which leads to lower overall accuracy ("Average"). Methods such as the 1vs1adjuster and post-hoc LA aim to enhance the accuracy of tail classes ("Few" classes) and improve the overall average accuracy.
>
> To clarify this further, we have added the following explanations to Appendix F.3 regarding the experimental settings and results:
> *"In addition to the overall average accuracy (Average), we also report the average accuracy for the 'Many,' 'Medium,' and 'Few' categories. Refer to Appendix F.1 for details on these categorizations."*
>
> *"In all results, the Baseline shows higher accuracy for the 'Many' category, as naive training on long-tailed data results in outputs biased toward head classes. The 1vs1adjuster and post-hoc LA methods successfully adjust for this bias by increasing the accuracy of 'Few' classes, thereby improving the overall average accuracy.
> As shown in Section 5.3, MLA achieves nearly the same average accuracy as the 1vs1 Adjuster across all datasets. Additionally, MLA outperforms ALA in terms of average accuracy for CIFAR100-LT."*
>
> > W3: The descriptions for tables and figures are too brief, making it difficult for readers to grasp the full context of the results. Adding more detailed captions would help in understanding the findings.
>
> A3: We have added explanations for the figures and tables. For instance, we have inserted the following sentence in the caption of Table 1:
> *"For models that achieve 100% training accuracy, $\gamma_\times^\*$ and $\gamma_{\mathrm{1v1}}^\*$ are slightly higher than 0.5."*
>
>
> > W4: The paper contains numerous formal definitions and formulas, which can disrupt the reading flow, especially for readers who are not deeply familiar with this specific research area.
>
> A4: To improve the flow of the paper, we have included several supplementary explanations. For example, at L189, we have added the motivation for introducing the Rademacher complexity as follows:
> *"To evaluate the generalization performance of $\mathcal{H}_l$, we use the Rademacher complexity."*
> Additionally, at L237, we have included an intuitive explanation of angular bound probability as follows:
> *"The angular bound probability can also be seen as concentration of the class $k$ features."*
>
>
> > Q1: Why did you primarily focus on ResNet models for the image analysis? Were there specific reasons for this choice over other architectures?
>
> A5: We selected ResNet as our primary architecture because it is a widely recognized benchmark in LTR. Numerous LTR studies, including ALA and MLA [1, 2, 3], have utilized ResNet for their experiments.
>
> Additionally, in line with [3], we conducted experiments using an MLP model on the Helena dataset (as detailed in Section 5.1). A similar performance was observed in this experiment. This indicates that our findings are not confined to a specific model architecture.

---

> ### Author Response · Authors · 2024-11-21
>
> > Q2:  Why did you create imbalanced versions of balanced datasets instead of using inherently imbalanced datasets? For example, datasets that exhibit natural class imbalances could offer more realistic evaluations.
>
> A6: We utilized CIFAR10/100-LT created from CIFAR10/100 because CIFAR10/100-LT are standard benchmark datasets in LTR. Many LTR studies, including ALA and MLA [1, 2, 3], have employed these datasets for their experiments.
>
> In addition, we followed [3] by incorporating experiments on the Helena dataset, which is inherently imbalanced with $\rho \simeq 40$, as outlined in Appendix F.1.3. The consistent results across both synthetic and naturally imbalanced settings underscore the effectiveness of our method in diverse scenarios.
>
> To highlight this, we added the following sentence to L399 in Section 5.1:
> *"We can also show the effectiveness of the method on real data using Helena because this is an inherently imbalanced dataset with $\rho \simeq 40$."*
>
>
> > Q3: Although the paper focuses on post-hoc LA methods, did you compare the results with loss-function-based approaches? How does MLA's performance compare to other logit adjustment methods?
>
> A7: This paper primarily centers on theoretical analysis, with the theoretical findings validated quantitatively through experiments. Comparing our method with other methods would not yield insights directly relevant to the theoretical contributions of this work. Including such comparisons might introduce experimental results that detract from the central narrative of our paper, potentially diminishing its coherence.
> It is uncommon in machine learning conferences to present experimental results beyond this type of quantitative validation. For example, several papers that focus on theoretical analysis without comprehensive comparisons [3, 4] have also been accepted at ICLR. Therefore, underrating this work on the grounds of not including such comparisons would compromise the fairness of the review process.
>
> Nonetheless, we conducted supplementary experiments to provide additional context and clarify the relationship with loss-function-based ALA as a highly relevant method. Specifically, we evaluated a loss-function-based ALA with $\gamma_+ = 1$ and compared its performance with post-hoc LA. All other experimental settings followed those described in Section 5.1.
>
> The results for CIFAR10/100-LT datasets are summarized below. As shown, the loss-function-based ALA achieves performance comparable to post-hoc ALA. This demonstrates that MLA achieves accuracy equivalent to, or better than, loss-function-based ALA.
>
>
> CIFAR100-LT
> |                  |    Loss ALA     |  Post-hoc ALA   |       Post-hoc MLA       |
> | ---------------- | :-------------: | :-------------: | :----------------------: |
> | Average Accuracy | $50.4_{\pm0.1}$ | $50.7_{\pm0.4}$ | $\mathbf{53.0_{\pm0.3}}$ |
>
> CIFAR10-LT
> |                  |         Loss ALA         |       Post-hoc ALA       |       Post-hoc MLA       |
> | ---------------- | :----------------------: | :----------------------: | :----------------------: |
> | Average Accuracy | $\mathbf{78.6_{\pm0.5}}$ | $\mathbf{79.7_{\pm1.1}}$ | $\mathbf{79.6_{\pm1.1}}$ |
>
>
>
> In making these revisions, we believe the paper has achieved a higher level of rigor and clarity. We hope these changes address your concerns and align with your expectations for an even more impactful contribution to the field. If you have any further suggestions or specific areas you believe could benefit from additional attention, we welcome your guidance. Your expertise is invaluable to us, and we are committed to delivering a paper that meets the highest standards. If there are no further corrections, we kindly request that you raise the score so that this paper can be accepted.
>
>
> [1] Long-tail learning via logit adjustment, Menon+, ICLR2021 Spotlight
> [2] Adjusting decision boundary for class imbalanced learning, Kim+, 2019
> [3] Exploring Weight Balancing on Long-Tailed Recognition Problem, Hasegawa+, ICLR2024
> [4] Improved Sample Complexities for Deep Neural Networks and Robust Classification via an All-Layer Margin, Wei+, ICLR 2020

---

> > ### Comment · Reviewer_K7Rv · 2024-11-25
> > **Thank you for the clarification**
> >
> > Thank you for addressing the concerns raised in the review. The clarifications and revisions provided sufficiently address the key points. I will maintain my score, leaning towards acceptance.

---

### Official Review · Reviewer_EiUb · 2024-11-04

**Soundness:** 2
**Presentation:** 3
**Contribution:** 2
**Rating:** 5
**Confidence:** 4

**Summary:**

The paper aims to provide a theoretical justification for Multiplicative Logit Adjustment (MLA) in long-tailed recognition by linking it to Neural Collapse theory. The authors develop a theoretical framework for optimal decision boundary adjustment based on feature spread estimates from Neural Collapse and demonstrate that MLA approximates this optimal method.

**Strengths:**

The framework that connects MLA with Neural Collapse is novel.  The assumptions and conditions are clearly stated, and the notation is consistent and well-defined.

**Weaknesses:**

1. **Motivation**:
   The paper fails to establish a compelling motivation for the study. The claim that "MLA has demonstrated significant empirical success" is not sufficiently substantiated. There is no systematic analysis of the current strengths and challenges of MLA in long-tailed recognition.

2. **Theory-Practice Gap**:
   The connection between optimal decision boundary adjustment and actual performance improvements in long-tailed scenarios is weak. While the paper seems to demonstrate the effectiveness of MLA as a close approximation to the 1vs1adjuster from both theoretical and experimental perspectives, and contrasts it with Additive Logit Adjustment (ALA), it does not thoroughly analyze how MLA addresses the specific challenges of long-tailed learning from a theoretical standpoint. Additionally, there is no comparison with state-of-the-art long-tailed baselines or discussion on how MLA could be integrated with them. The experimental section focuses more on hyperparameter tuning for MLA rather than examining its impact on long-tailed learning.

**Questions:**

The paper mentions that "MLA has demonstrated significant empirical success," but it does not provide sufficient evidence to support this claim. Could more literature or experimental results be added to demonstrate the current advantages of MLA in long-tailed recognition?

The theoretical comparison between MLA and ALA is addressed in some aspects, but why is there no comparison with other advanced baselines in long-tailed recognition (such as the latest contrastive learning methods or distribution-balanced loss)?

---

> ### Author Response · Authors · 2024-11-18
>
> We sincerely appreciate you for your valuable time and constructive feedback. We would like to address some of the points raised in your review and provide additional context or clarification that we believe could contribute to your acceptance of our paper. Revised sections are indicated in red in the PDF.
>
>
>  Before addressing each specific point, we would like to reiterate that our research centers on the theoretical analysis of post-hoc logit adjustment (LA) within the framework of long-tailed recognition (LTR). This method is particularly advantageous as it requires no specialized training; it only modifies the logits during inference on a conventionally trained model to enhance accuracy. Despite its straightforward implementation and low computational cost, this approach yields significant accuracy improvements, making it highly applicable to real-world scenarios. Notably, additive LA (ALA) [1] was featured as a **Spotlight paper at ICLR 2021** and has received over 700 citations. Thus, we believe that **theoretical contributions in this area hold substantial potential to advance the field of LTR**.
>
> We would like to address the key points:
>
> > W1: Motivation: The paper fails to establish a compelling motivation for the study. The claim that "MLA has demonstrated significant empirical success" is not sufficiently substantiated. There is no systematic analysis of the current strengths and challenges of MLA in long-tailed recognition.
>
> A1: Multiplicative LA (MLA) has been confirmed effective in various studies. First, MLA shows performance equivalent to or better than considerably influential ALA. Comparisons between MLA and ALA have been validated by [2] and our experiments on various datasets (see Section 5.3). Second, [6] has shown that the regions occupied by features for each class differ significantly in LTR, which supports the effectiveness of class-dependent temperatures, a method equivalent to MLA.
> Third, [7] and [8] indicate the efficacy of techniques similar to MLA beyond image classification. For instance, in object detection, MLA based on the frequency of object occurrences, known as inverse image frequency, has been proposed [7]. Similarly, in text retrieval tasks, a widely used method called inverse document frequency [8] adjusts the score function by multiplying it by the inverse of the overall word frequency. These suggest that MLA is not confined to classification in LTR but is a versatile technique applicable across various domains.
>
> However, MLA has lacked a rigorous theoretical analysis until now. Our study addresses that gap by providing a theoretical guarantee of MLA optimality, thereby contributing significantly to the LTR field.
>
> To better convey this, we have included the following explanations of the existing work in Appendix B1.
>
> *“MLA has been validated for its effectiveness across various studies. Hasegawa & Sato (2023) demonstrated that MLA performs on par with or better than ALA under realistic conditions. Additionally, Ye et al. (2022) highlighted significant differences in the feature space occupied by each class in LTR, confirming the effectiveness of class-dependent temperatures, which is equivalent to MLA. Beyond image classification, various studies suggest the utility of MLA-like adjustments in other fields. For instance, in object detection, Alexandridis et al. (2023) proposed an MLA variant based on object occurrence frequency, called inverse image frequency. Similarly, in text retrieval tasks, the inverse document frequency  (Salton & Buckley, 1988), which adjusts the score function by multiplying the inverse frequency of words in a document, is widely used. Thus, MLA is not only applicable to LTR in classification problems but also serves as a fundamental technique with potential applications across a broad range of domains.”*

---

> > ### Author Response · Authors · 2024-11-18
> >
> > > W2-1:The connection between optimal decision boundary adjustment and actual performance improvements in long-tailed scenarios is weak. While the paper seems to demonstrate the effectiveness of MLA as a close approximation to the 1vs1adjuster from both theoretical and experimental perspectives, and contrasts it with Additive Logit Adjustment (ALA), it does not thoroughly analyze how MLA addresses the specific challenges of long-tailed learning from a theoretical standpoint.
> >
> > A2-1: Section 4 demonstrates how the MLA effectively solves the specific problem of LTR using a two-stage theory. 1vs1Adjuster addresses specific challenges in LTR, and since MLA serves as an approximation of this method, it similarly resolves these issues. When the sample size per class is highly imbalanced, as is often the case in LTR, the feature space occupied by each class can vary significantly (as explained in L209\~L255 and Proposition 1). To address this, it is possible to adjust the decision boundary to maximize the average accuracy across all classes (as explained in L256\~L297 and Propositions 2 and 3). 1vs1Adjuster implements these adjustments. Section 4.3 demonstrates that MLA approximates the 1vs1Adjuster sufficiently, even within the LTR setting. Thus, MLA effectively addresses the challenges of LTR, making it a valuable approach.
> >
> > To clarify this point, we have added the following sentence at the end of the second paragraph in Section 4.2.
> >
> > *"In LTR, sample sizes vary significantly across classes, resulting in a considerable imbalance in the angular deviation. Thus, we aim to adjust the decision boundary accordingly."*
> >
> >
> >
> > > W2-2: Additionally, there is no comparison with state-of-the-art long-tailed baselines or discussion on how MLA could be integrated with them. The experimental section focuses more on hyperparameter tuning for MLA rather than examining its impact on long-tailed learning.
> >
> > A2-2: This paper focuses on theoretical analysis, and the validity of the theoretical results is quantitatively evaluated through experiments. Even if we were to conduct comparative experiments with SOTA methods, the theoretical analysis does not provide any insights or implications regarding the relationship with SOTA. As a result, including such experimental results would introduce findings unrelated to the paper's narrative, ultimately undermining the consistency of its storyline. It is uncommon in machine learning conferences to present experimental results beyond this type of quantitative validation. For example, several papers that focus on theoretical analysis without SOTA comparisons [2], [3], [9] have also been accepted at ICLR. Thus, rejecting our work on this basis could compromise the fairness of the review process.
> >
> > However, since this theoretical analysis is a general framework that does not interfere with the combination with other methods, verifying whether it is similarly effective when combined with other approaches aligns with the purpose of the paper. MLA can be easily combined with other methods by simply modifying the logit operations during the inference of trained models. According to [4], LTR methods can be categorized into three groups: “Information Augmentation,” “Module Improvement,” and “Class Re-balancing.” Post-hoc LA can be combined with methods from the first two categories to enhance performance further. For example, DODA [5], which falls under “Information Augmentation,” can be combined with MLA to improve accuracy.
> >
> > Below, we compare the average accuracy on CIFAR100-LT of models trained with cross-entropy loss and DODA, both with and without the application of MLA. Attention that this experiment is based on the implementation and settings from [5], making it not directly comparable to our other experimental results. However, these findings demonstrate that MLA can be seamlessly integrated with other LTR methods, yielding further accuracy gains. We have included this analysis in Appendix F.4 of our paper.
> >
> >
> > |                  | CE+DODA | CE+DODA+MLA |
> > | ---------------- | :-----: | :---------: |
> > | Average accuracy |  44.9   |    **49.1**     |

---

> > > ### Author Response · Authors · 2024-11-18
> > >
> > > > Q1: The paper mentions that "MLA has demonstrated significant empirical success," but it does not provide sufficient evidence to support this claim. Could more literature or experimental results be added to demonstrate the current advantages of MLA in long-tailed recognition?
> > >
> > > A3: See A1. We have added explanations in Appendix B.1 for [2], [6], [7] and [8], which provide evidence supporting the effectiveness of MLA.
> > >
> > >
> > > > Q2: The theoretical comparison between MLA and ALA is addressed in some aspects, but why is there no comparison with other advanced baselines in long-tailed recognition (such as the latest contrastive learning methods or distribution-balanced loss)?
> > >
> > > A4: See A2-2. This paper does not propose a new method but rather provides a theoretical analysis of existing methods. Therefore, the purpose of the experiments is to validate the theory quantitatively. Conducting additional experiments without theoretical justification would be unwarranted and lack relevance. In machine learning conferences, it is uncommon to include experimental results that extend beyond such quantitative validations. For instance, [2], [3] and [9] have been accepted without SOTA comparisons in ICLR. Therefore, a rejection of our work on this basis could compromise the fairness of the review process.
> > >
> > > We believe that these revisions contribute positively to the overall strength of our paper, and we hope they align with your expectations. We sincerely appreciate the time and effort you dedicated to the review process. If you have any additional suggestions or comments to ensure the paper's excellence, we are more than willing to incorporate them. If there are no further corrections, we kindly request that you raise the score so that this paper can be accepted. Thank you once again for your generous evaluation and constructive feedback. We are excited about the opportunity to contribute this work to the academic community.
> > >
> > >
> > > [1] Long-tail learning via logit adjustment, Menon+, ICLR2021 Spotlight
> > > [2] Exploring Weight Balancing on Long-Tailed Recognition Problem, Hasegawa+, ICLR2024
> > > [3] Improved Sample Complexities for Deep Neural Networks and Robust Classification via an All-Layer Margin, Wei+, ICLR 2020
> > > [4] Deep Long-Tailed Learning: A Survey, Zhang+, 2021
> > > [5] Kill Two Birds with One Stone: Rethinking Data Augmentation for Deep Long-tailed Learning, Wang+, ICLR2024
> > > [6] Identifying and Compensating for Feature Deviation in Imbalanced Deep Learning, Ye+, 2020
> > > [7] Inverse Image Frequency for Long-tailed Image Recognition, Alexandridis+, IEEE Transactions on Image Processing, 2023
> > > [8] Term-weighting approaches in automatic text retrieval. Salton+, Information Processing & Management, 1988
> > > [9] Understanding Why Generalized Reweighting Does Not Improve Over ERM, Zhai+, ICLR2023

---

> > > > ### Author Response · Authors · 2024-11-21
> > > >
> > > > Regarding A2-1, the emphasis in Section 4.3 is not on highlighting that MLA approximates 1vs1adjuster but rather on demonstrating that MLA effectively addresses the LTR problem. Directly analyzing MLA proved challenging; therefore, we illustrated its effectiveness through 1vs1adjuster, which has the optimal decision boundaries for LTR. Consequently, 1vs1adjuster is not a proposed method but a tool to substantiate the optimality of MLA. The concept of using this intermediary method itself represents one of our contributions.
> > > >
> > > > To clarify this point further, we have made the following changes:
> > > > - Revised the second contribution in the Introduction to:
> > > > *”We demonstrate that MLA is effective for LTR by proving it has similar decision boundaries to the method based on the aforementioned theory (Section 4.3). This clarifies under what conditions this approximation holds and how adjustments should be made."*
> > > > - Changed the title of Section 4.3 to *"MLA solves LTR problems with decision boundaries akin to 1vs1adjuster"*.
> > > > - Revised the first two sentences of Section 4.3 as follows:
> > > > *"We demonstrate that MLA addresses the LTR problem. Since Section 4.2 proves 1vs1adjuster has the optimal decision boundaries for LTR, we verify MLA operates similarly to 1vs1adjuster.
> > > > "*
> > > >
> > > > We are grateful for your feedback, which has helped us to enhance the clarity of our contribution. We thank you for your assistance.

---

> > > > > ### Author Response · Authors · 2024-12-02
> > > > >
> > > > > Dear Reviewer EiUb,
> > > > >
> > > > > We sincerely thank you for your thorough and constructive feedback on our paper. Your detailed comments have been invaluable in helping us refine and enhance the quality of our research.
> > > > >
> > > > > In this revised manuscript, we have carefully addressed each concern and question raised in the previous review. The revisions directly respond to your suggestions, significantly improving the paper's scientific rigor and clarity.
> > > > >
> > > > > We deeply value the iterative dialogue between authors and reviewers, as crucial in advancing scientific knowledge. We believe the current version of our manuscript reflects our commitment to maintaining the highest standards of academic research.
> > > > >
> > > > > We have meticulously addressed each critique, and we are confident that our revisions substantively improve the overall quality of the paper. If no further substantive modifications are required, we would appreciate the opportunity to have our paper's score reconsidered for acceptance.
> > > > >
> > > > > Again, thank you for your time, expertise, and valuable insights throughout this review.
> > > > >
> > > > > Sincerely,
> > > > > Authors

---

> > > > > > ### Comment · Reviewer_EiUb · 2024-12-02
> > > > > > **Thank you for the clarification**
> > > > > >
> > > > > > Thank you for your detailed responses and the additional clarifications provided. I appreciate the effort you have made to address my concerns, particularly regarding the motivation of the study. Your explanations have helped clarify some of my earlier doubts on this aspect.
> > > > > >
> > > > > > However, I find the arguments such as “For instance, [2], [3] and [9] have been accepted without SOTA comparisons in ICLR. Therefore, a rejection of our work on this basis could compromise the fairness of the review process.” and “The larger number of classes and samples per class align closely with the conditions assumed in our theory, making additional experiments less critical.” less convincing in addressing the core concerns raised by myself and other reviewers regarding the robustness and applicability of the theoretical contributions in this work. After careful consideration, I have decided to cautiously raise my evaluation to a 5.

---

> ### Author Response · Authors · 2024-12-03
>
> Thank you for your reply and the revised score. We apologize for any misunderstanding caused by our previous explanations. To address the remaining concerns, we provide detailed responses below:
>
> > W1: “For instance, [2], [3] and [9] have been accepted without SOTA comparisons in ICLR. Therefore, a rejection of our work on this basis could compromise the fairness of the review process.” ~ less convincing in addressing the core concerns raised by myself and other reviewers regarding the robustness and applicability of the theoretical contributions in this work.
>
> A1: We emphasize that comparisons with methods outside the scope of post-hoc LA are not directly relevant to this work. Our study is a theoretical analysis specifically within the post-hoc LA framework, and we conducted experiments to validate the theoretical findings. Accuracy comparisons with unrelated baseline or SOTA methods fall outside the theoretical scope of this work and would lack justification.
>
> Moreover:
> 1. Comparisons with other baseline methods have already been thoroughly covered in the original paper of MLA [1], making further redundancy unnecessary.
> 2. Techniques such as contrastive learning, which require extended convergence times, are not feasible to evaluate within the time constraints of this review process.
>
> However, we have already conducted additional experiments with loss-function-based baseline methods closely aligned with post-hoc LA. These results include comparisons between the loss-function version of ALA and various post-hoc LA methods. We show the results for CIFAR10/100-LT again in the table below. As shown, the loss-function-based ALA achieves performance comparable to post-hoc ALA. This demonstrates that MLA achieves accuracy equivalent to, or better than, loss-function-based ALA.
>
> CIFAR100-LT
> |                  |    Loss ALA     |  Post-hoc ALA   |       Post-hoc MLA       |
> | ---------------- | :-------------: | :-------------: | :----------------------: |
> | Average Accuracy | $50.4_{\pm0.1}$ | $50.7_{\pm0.4}$ | $\mathbf{53.0_{\pm0.3}}$ |
>
>
> CIFAR10-LT
>
> |                  |         Loss ALA         |       Post-hoc ALA       |       Post-hoc MLA       |
> | ---------------- | :----------------------: | :----------------------: | :----------------------: |
> | Average Accuracy | $\mathbf{78.6_{\pm0.5}}$ | $\mathbf{79.7_{\pm1.1}}$ | $\mathbf{79.6_{\pm1.1}}$ |
>
>
> > W2: “The larger number of classes and samples per class align closely with the conditions assumed in our theory, making additional experiments less critical.” less convincing in addressing the core concerns raised by ~  other reviewers regarding the robustness and applicability of the theoretical contributions in this work
>
> A2: Our theoretical analysis assumes a sufficiently large number of classes (addressed in Section 4.3) and a sufficiently large sample size per class (addressed in Section 4.2). These assumptions are inherently satisfied in large-scale datasets, which further enhance the applicability of our theoretical results in such settings.
>
> Additionally, we have already addressed other concerns raised by the other reviewers. For instance, we have clarified that our theoretical framework is model-agnostic and remains valid even for naturally imbalanced datasets.
>
> We deeply appreciate your continued attention to these issues. If the remaining concerns have now been resolved, we kindly request a further score adjustment to reflect this resolution. Should any additional questions remain, please do not hesitate to contact us.
>
> [1] Adjusting decision boundary for class imbalanced learning, Kim+, 2019

---

### Official Review · Reviewer_Hejv · 2024-11-04

**Soundness:** 3
**Presentation:** 3
**Contribution:** 3
**Rating:** 6
**Confidence:** 3

**Summary:**

This paper provides a theoretical foundation for the Multiplicative Logit Adjustment (MLA) method used in long-tailed recognition tasks. First, the authors develop a theory for optimally adjusting decision boundaries by leveraging feature spread estimates derived from neural collapse (NC). They then demonstrate that MLA effectively approximates this optimal adjustment method. Experiments conducted on various long-tailed datasets validate the practical applicability of this approximation, showing that MLA performs well under realistic conditions.

**Strengths:**

- This paper presents a theory for optimally adjusting decision boundaries based on NC and links it to the MLA method, providing a strong theoretical explanation for MLA and adding depth to its empirical success.
- The authors conduct a series of experiments on long-tailed datasets to validate the theory and demonstrate the practical utility of MLA.
- The authors offer both theoretical and empirical comparisons between MLA and ALA, clarifying their differences and highlighting the advantages of MLA.

**Weaknesses:**

- The iNaturalist dataset is a widely used large-scale long-tailed dataset. Conducting experiments on this dataset would be beneficial to examine the behavior of MLA, ALA, and the 1vs1 adjuster in a larger-scale real-world context. Additionally, testing on CIFAR-LT with more extreme imbalance ratios would further evaluate the effectiveness of the proposed method.

- The theoretical framework relies on some strict assumptions that may not always hold in practice, such as the assumption of conditions under which NC occurs, which may not be valid in highly imbalanced real-world datasets. Similarly, the assumption that $\psi \rightarrow \pi/2$ does not hold when $K$ is small.
- There are some typos, such as on line 373, where "the optimal decision boundaries during inference is effected by" should be corrected to "the optimal decision boundaries during inference are affected by".

**Questions:**

Please refer to the weakness section.

---

> ### Author Response · Authors · 2024-11-21
>
> We sincerely appreciate your time and thoughtful evaluation of our paper. Your positive feedback is genuinely encouraging, and we are grateful for the constructive comments that have undoubtedly contributed to improving our work. While we understand and value the favorable assessment, we would like to address a few points raised in your review where I believe additional context or clarification could potentially enhance the overall evaluation of our paper. Revised sections are indicated in red in the PDF.
>
> > W1: The iNaturalist dataset is a widely used large-scale long-tailed dataset. Conducting experiments on this dataset would be beneficial in examining the behavior of MLA, ALA, and the 1vs1 adjuster in a larger-scale real-world context. Additionally, testing on CIFAR-LT with more extreme imbalance ratios would further evaluate the effectiveness of the proposed method.
>
> A1: We have conducted additional experiments on a more imbalanced CIFAR-LT dataset ($\rho=200$) and included the results in Appendix F.5. All other experimental settings were identical to those described in Section 5.1. Due to space constraints, we present only the average accuracy after hyperparameter tuning on CIFAR100-LT here. Please refer to Appendix F.5 for the comprehensive results. Similar to the case with $\rho=100$, the results demonstrate that MLA effectively approximates the 1vs1adjuster in all scenarios. MLA achieves higher mean accuracy than ALA on CIFAR100-LT in this case as well.
>
>
> |                  |    Baseline     |       ALA       |           1v1            |           MLA            |
> | ---------------- | :-------------: | :-------------: | :----------------------: | :----------------------: |
> | Average accuracy | $43.1_{\pm0.3}$ | $45.7_{\pm0.4}$ | $\mathbf{47.8_{\pm0.4}}$ | $\mathbf{47.7_{\pm0.5}}$ |
>
> Regarding experiments on iNaturalist, we could not conduct these during the rebuttal period for the following three reasons. First, the computational cost in our environment is prohibitively high. For instance, a typical LTR experiment on iNaturalist requires training with a batch size of 500, which demands substantial VRAM or extensive GPU hours [1]. Second, on such large-scale datasets, the theoretical assumptions underlying our work are more likely to hold, reducing the necessity for these experiments. The larger number of classes and samples per class align closely with the conditions assumed in our theory, making additional experiments less critical. Finally, other LTR papers without experiments on iNaturalist [2], [3] have also been accepted at ICLR. For these reasons, including these experiments is unnecessary to achieve a high evaluation.
>
>
> > W2: The theoretical framework relies on some strict assumptions that may not always hold in practice, such as the assumption of conditions under which NC occurs, which may not be valid in highly imbalanced real-world datasets. Similarly, the assumption that ψ→π/2 does not hold when K is small.
>
> A2: The experimental results suggest that MLA effectively approximates the 1vs1Adjuster even under non-ideal conditions. In the experiments described in Appendix F.2, MLA and the 1vs1Adjuster exhibit similar decision boundaries, even in scenarios where NC does not fully hold (e.g., Helena and ImageNet-LT). Furthermore, Section 5.3 demonstrates that MLA performs nearly identically to the 1vs1Adjuster in terms of accuracy, even in cases with a small number of classes, such as CIFAR10-LT, indicating that these conditions do not pose practical issues.
>
> Additionally, we consider these conditions to be of relatively low importance. By definition, "long"-tailed recognition is characterized by scenarios involving a large number of classes [4]. Moreover, some methods like ETF Classifier [5] promote NC even in imbalanced data settings (as explained in L112–L115). Considering these factors, we believe that the assumptions we made are reasonable.
>
> To better convey this point, we revised the sentence starting from "In LTR" in L36 as follows:
> *"In LTR, there are many classes, and model predictions are often biased toward head classes."*
>
> Additionally, we added the following sentence to L278:
> *"This is a reasonable assumption in LTR (Yang et al., 2022a)."*
>
>
> > W3: There are some typos, such as on line 373, where "the optimal decision boundaries during inference is effected by" should be corrected to "the optimal decision boundaries during inference are affected by".
>
> A3: We have corrected the points you pointed out. Thank you for your comments.

---

> > ### Author Response · Authors · 2024-11-21
> >
> > In responding to your feedback, our goal is to not only meet the criteria for acceptance but also to exceed expectations where possible. We believe that by addressing these points, we can enhance the overall contribution and impact of our paper. We genuinely appreciate your careful consideration of our work, and we are committed to ensuring that the revised manuscript reflects the highest standards of quality and clarity. If you have any additional suggestions or concerns, please don't hesitate to let us know. If there are no further corrections, we kindly request that you raise the score so that this paper can be accepted.
> >
> >
> > [1] Long-Tailed Recognition via Weight Balancing, Alshammari+, CVPR2022
> > [2] Heteroskedastic and Imbalanced Deep Learning with Adaptive Regularization, Cao+, ICLR 2021
> > [3] Exploring Weight Balancing on Long-Tailed Recognition Problem, Hasegawa+, ICLR2024
> > [4] A Survey on Long-Tailed Visual Recognition, Yang+, International Journal of Computer Vision, 2022
> > [5] Inducing Neural Collapse in Imbalanced Learning: Do We Really Need a Learnable Classifier at the End of Deep Neural Network?, Yang+, NeurIPS2022

---

> > > ### Author Response · Authors · 2024-12-02
> > >
> > > Dear Reviewer Hejv,
> > >
> > > We sincerely thank you for your thoughtful and constructive feedback on our manuscript. Your detailed comments have been invaluable in helping us refine and enhance the quality of our research.
> > >
> > > We have carefully addressed each concern and question raised in your previous review. Our point-by-point response demonstrates our commitment to thoroughly engaging with your insights and improving the scientific rigor of our work. These revisions have significantly strengthened the manuscript’s clarity, methodology, and overall contribution to the field.
> > >
> > > We are grateful for your meticulous examination, which has enabled us to evaluate and enhance our research. The collaborative nature of the peer review process is essential for advancing scientific understanding, and we deeply value this opportunity to engage in constructive dialogue.
> > >
> > > Having comprehensively addressed all previous comments, we kindly request a final review of our revised manuscript. If no further substantive changes are required, we would greatly appreciate the opportunity to have the manuscript’s score reconsidered for acceptance.
> > >
> > > We remain dedicated to producing high-quality research and are eager to finalize this manuscript for publication.
> > >
> > > Thank you once again for your continued support and guidance.
> > >
> > > Sincerely, Authors

---

### Meta-Review · Area_Chair_iHXF · 2024-12-23

**Metareview:**

This paper studies the multiplicative logit adjustment (MLA) method proposed recently for the long-tail recognition (LTR) problem. The paper formally shows that MLA can be seen as an approximation of a theoretically-guided approach based on a neural collapse (NC) based adjusting of the optimal decision boundaries. The theoretical analysis also results in some practical benefits, such as better performance in more realistic LTR settings and hyperparameter tuning.

The paper received generally positive reviews from the reviewers. There were some concerns such as (1) lack of experiments on very large-scale LTR datasets such as iNaturalist (Reviewer Hejv), and (2) concerns about some of the theoretical assumptions (Reviewer Hejv), (3) motivation and theory-practice gap (Reviewer EiUb). There were some additional concerns that the experiments are not strongly demonstrating the effectiveness of MLA but are rather focused more on how it guides hyperparameter tuning

The authors provided a detailed response to these concerns. In the end, while two reviewers leaned towards acceptance with scores of 6, Reviewer EiUb remained a bit cautious with a borderline score of 5.

Considering the reviews, the author rebuttal, the discussion, and my own reading of the paper, my assessment is that the paper does offer a nice understanding/justification of MLA, along with some practical guidelines on how to use it effectively in realistic real-world LTR problems. I therefore recommend the paper for acceptance. At the same time, the authors should incorporate the reviewer feedback in the final version.

**Additional Comments On Reviewer Discussion:**

All the reviewers were in general positive about the paper. The rebuttal was considered. In the end, while two reviewers leaned towards acceptance with scores of 6, Reviewer EiUb remained a bit cautious with a borderline score of 5. However, the reviewer's concerns were not very critical and the other positives of the paper outweigh those concerns.

---

### Decision · Program_Chairs · 2025-01-22

Accept (Poster)